# Epithelial cell chirality emerges through the dynamic concentric pattern of actomyosin cytoskeleton

**Takaki Yamamoto[1,2†], Tomoki Ishibashi[1†], Yuko Mimori-Kiyosue[3], Sylvain Hiver[4], Naoko Tokushige[1,3], Mitsusuke Tarama[1,5], Masatoshi Takeichi[4], Tatsuo Shibata[1*]**

[1]Laboratory for Physical Biology, RIKEN Center for Biosystems Dynamics Research, Kobe, Japan; [2]Nonequilibrium Physics of Living Matter RIKEN Hakubi Research Team, RIKEN Center for Biosystems Dynamics Research, Kobe, Japan; [3]Laboratory for Molecular and Cellular Dynamics, RIKEN Center for Biosystems Dynamics Research, Kobe, Japan; [4]Laboratory for Cell Adhesion and Tissue Patterning, RIKEN Center for Biosystems Dynamics Research, Kobe, Japan; [5]Department of Physics, Kyushu University, Fukuoka, Japan

**\*For correspondence:**
tatsuo.shibata@riken.jp

[†]These authors contributed equally to this work

**Competing interest:** The authors declare that no competing interests exist.

**Abstract** The chirality of tissues and organs is essential for their proper function and development. Tissue-level chirality derives from the chirality of individual cells that comprise the tissue, and cellular chirality is considered to emerge through the organization of chiral molecules within the cell. However, the principle of how molecular chirality leads to cellular chirality remains unresolved. To address this fundamental question, we experimentally studied the chiral behaviors of isolated epithelial cells derived from a carcinoma line and developed a theoretical understanding of how their behaviors arise from molecular-level chirality. We first found that the nucleus undergoes clockwise rotation, accompanied by robust cytoplasmic circulation in the same direction. During the rotation, actin and Myosin IIA assemble into the stress fibers with a vortex-like chiral orientation at the ventral side of the cell periphery, concurrently forming a concentric pattern at the dorsal side. Further analysis revealed that the intracellular rotation is driven by the concentric actomyosin filaments located dorsally, not by the ventral vortex-like chiral stress fibers. To elucidate how these concentric actomyosin filaments induce chiral rotation, we analyzed a theoretical model developed based on the theory of active chiral fluid. This model demonstrated that the observed cell-scale unidirectional rotation is driven by the molecular-scale chirality of actomyosin filaments even in the absence of cell-scale chiral orientational order. Our study thus provides novel mechanistic insights into how the molecular chirality is organized into the cellular chirality, representing an important step toward understanding left–right symmetry breaking in tissues and organs.

## Editor's evaluation

Although the actomyosin cytoskeleton has been shown to play an important role, the principles by which molecular chirality leads to the chirality of cells, tissues, and organs remain largely unexplored. This important study reveals that the concentric actomyosin network at the apical side of Caco-2 cells, rather than the ventral chiral stress fibers, drives the rotational movement of the nucleus and cytoplasmic flow in the same direction. The convincing data are supported by a theoretical model based on the theory of active fluids, which explains how unidirectional rotation at the cellular scale can arise from the chirality of actomyosin filaments at the molecular scale, even in the absence of chiral orientational order at the cellular scale.

## Introduction

Left–right asymmetry is ubiquitously observed in the bodies and organs of organisms. Despite extensive research, however, we still do not have a complete understanding of how left–right asymmetric structures are formed at an organismal scale. The breaking of left–right symmetry at the body and organ scale has been investigated in embryonic bodies, such as early vertebrate embryo (*Hamada and Tam, 2014*; *Blum and Ott, 2018*), nematodes (*Naganathan et al., 2014*; *Pimpale et al., 2020*; *Sugioka and Bowerman, 2018*), and pond snails (*Shibazaki et al., 2004*; *Davison et al., 2016*; *Abe and Kuroda, 2019*); and in organogenesis, such as embryonic hindgut (*Hozumi et al., 2006*; *Taniguchi et al., 2011*; *Hatori et al., 2014*) and male genitalia (*Sato et al., 2015a*) in *Drosophila* and heart-looping in the chicken (*Ray et al., 2018*). Interestingly, in most of these cases, the left–right symmetry breaking at the organ scale is associated with chiral features at the cellular scale, indicating that cell-level chirality induces multicellular chirality (*Ishibashi et al., 2019*). Chiral dynamics have been observed in isolated single cells, such as nerve cells (*Tamada et al., 2010*), zebrafish melanophores (*Yamanaka and Kondo, 2015*), human foreskin fibroblasts (HFF) (*Tee et al., 2015*; *Tee et al., 2023*), and Madin–Darby canine kidney (MDCK) cells (*Chin et al., 2018*). Furthermore, experimental and theoretical studies have revealed that cell-intrinsic chirality drives left–right asymmetric morphogenesis of tissues (*Chen et al., 2012*; *Sato et al., 2015b*; *Yamamoto et al., 2020*) and organs (*Ray et al., 2018*). Therefore, to elucidate the mechanism of left–right symmetry breaking of organismal structures, it is essential to comprehensively investigate the mechanism underlying chiral dynamics at the single-cell scale.

In cells, there are many chiral components, such as amino acids, proteins, and DNA, and their proper organization can induce chiral properties of cells (*Brown and Wolpert, 1990*). Particularly, cytoskeletal molecules such as actin and microtubules have been suggested as candidate apparatuses driving chiral dynamics at the single-cell scale. For instance, actin and myosin are responsible for the chiral nuclear rotation of zebrafish melanophore (*Yamanaka and Kondo, 2015*), and the chiral neurite extension in nerve cells (*Tamada et al., 2010*). Unconventional Myosin 1D, in particular, plays an important role in the chiral morphogenesis in several species, including *Drosophila* (*Hozumi et al., 2006*; *Spéder et al., 2006*), zebrafish (*Juan et al., 2018*), and *Xenopus* (*Tingler et al., 2018*). Overexpression of this molecule can even induce chiral twisting in otherwise non-chiral organs in *Drosophila* (*Lebreton et al., 2018*). Actin cytoskeleton-related protein formins have also been shown to contribute to the chiral patterning of actin cytoskeleton in HFF (*Tee et al., 2015*; *Tee et al., 2023*). Formins are implicated in chiral morphogenesis in several organisms, including the snail chiral morphogenesis (*Davison et al., 2016*; *Kuroda et al., 2016*; *Abe and Kuroda, 2019*), *Drosophila* hindgut and genitalia chirality (*Chougule et al., 2020*), and chiral cortical flow in *C. elegans* (*Middelkoop et al., 2021*). Compared to the well-documented role of actomyosins, the involvement of microtubules in cellular chirality is less frequently reported. Nevertheless, microtubules have been shown to contribute to chirality in cultured human neutrophils (*Xu et al., 2007*). These studies attribute the chiral cell dynamics to the chiral rotating dynamics of actin and microtubules driven by molecular motors (*Nishizaka et al., 1993*; *Sase et al., 1997*). However, a mechanistic understanding of how these molecules generate cell-scale chirality is still not complete. Several attempts to gain mechanistic insights using theoretical models indicate that the chiral symmetry breaking at the cellular level requires spatial coordination of chiral cytoskeletal molecules (*Naganathan et al., 2014*; *Tee et al., 2015*). In particular, for the chirality of HFF, it has been proposed that the transverse actin fibers physically interact with radial actin fibers, which are screwed by formin, to drive the nuclear rotation in the counterclockwise direction (*Tee et al., 2015*; *Tee et al., 2023*). In the *C. elegans* embryo, it has been proposed that the actomyosin cortex generates active chiral torque and its spatial gradient induces the chiral symmetry breaking (*Naganathan et al., 2014*). Therefore, to crack the code of cellular chirality, it is important to elucidate how molecular-scale chiral activity spatially coordinates to trigger cellular-scale chirality.

In the present study, we investigated the behavior of Caco-2 cells, a typical epithelial cell line that was derived from colorectal adenocarcinoma. We found that, when these cells were singly isolated and cultured on substrates, the nucleus rotates along with the circulation of the cytoplasm in a clockwise direction, as viewed from above. We then showed that actin and Myosin II are responsible for this rotation of intracellular components. These cytoskeletal molecules formed concentric actomyosin filaments at the dorsal side of the cells, while they were organized into stress fibers with a vortex-like chiral orientation at the ventral side. Our experiments suggest that the former structure most likely

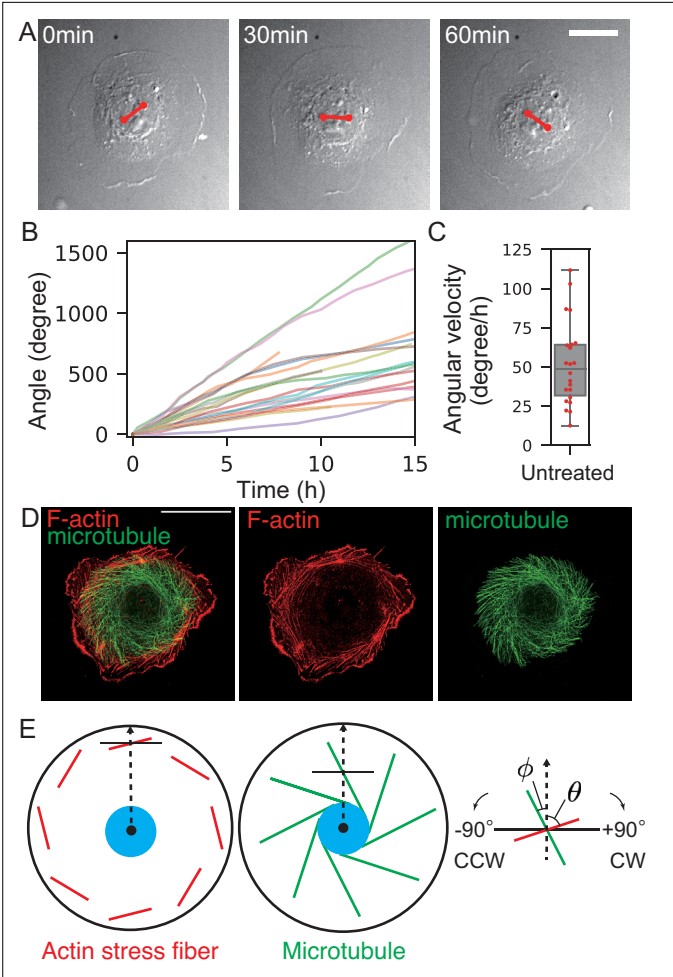

**Figure 1.** Chiral nuclear rotation and cytoplasmic flow in singly isolated Caco-2 cells. (**A**) Rotating nucleus probed by the rotation of nuclear texture. The endpoints of the red line segments are the positions of tracked landmarks of the nucleus. (**B**) The cumulative angle of nuclear rotation plotted against time and (**C**) average angular velocity averaged over the first 10 hr ($n = 22$). Here, positive angle values indicate clockwise rotation. (**D**) Chiral cytoskeletal structure of F-actin (phalloidin) and microtubule (immunostaining). Scale bar: 20 μm. (**E**) Schematic diagram of the orientation of actin stress fibers and microtubule.

The online version of this article includes the following figure supplement(s) for figure 1:

**Figure supplement 1.** Effect of physical environment on the nuclear rotation.

**Figure supplement 2.** Actin bundles in the peripheral region of the cell.

drives the rotating motion, implying that a cell nucleus can rotate without any cell-scale chiral orientational order of the cytoskeleton. To elucidate whether the concentric achiral pattern of the actomyosin filaments can indeed generate rotational flow, we analyzed a hydrodynamic model, based on the active chiral fluid theory, of a three-dimensional (3D) cell, considering the effect of molecular chirality of actin and myosin. We found that the concentric achiral structure of actomyosin can generate chiral cytoplasmic circulation, due to the force which originates from the molecular chirality of individual cytoskeletal components, even without cell-scale chiral structures. On the other hand, we found no evidence that radial actin fibers are involved in the nuclear rotation in Caco-2 cells, suggesting that there might be cell type-specific mechanisms to rotate cytoplasmic components.

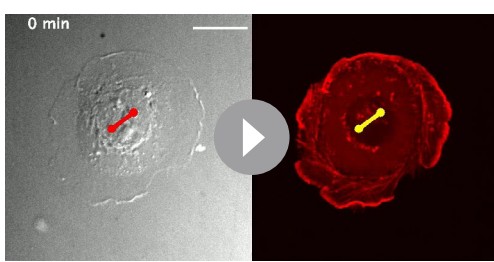

**Video 1.** Differential interference contrast (DIC) and fluorescent images of Caco-2 expressing Lifeact-RFP. Scale bar: 20 µm.

https://elifesciences.org/articles/102296/figures#video1

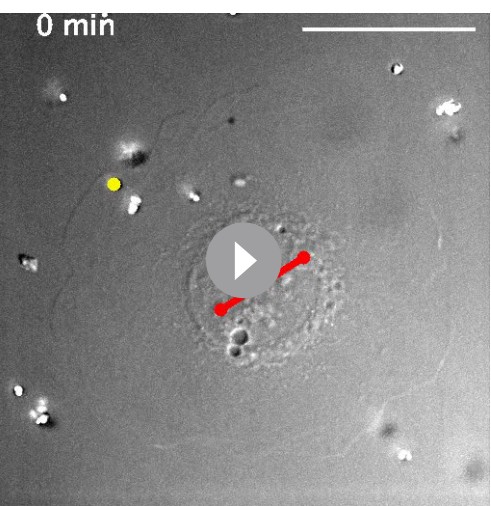

**Video 2.** Differential interference contrast (DIC) images of Caco-2 with beads attached to the dorsal membrane. Scale bar: 40 µm.

https://elifesciences.org/articles/102296/figures#video2

## Results

### Nuclei of singly isolated Caco-2 cells rotate in a clockwise direction

To study the rotational dynamics of epithelial cells, we cultured singly isolated Caco-2 cells and imaged them using a differential interference contrast (DIC) microscope (*Figure 1A*, *Video 1*). 76% of isolated Caco-2 cells spread circularly on a collagen-coated glass substrate, generating lamellipodia in all directions along the cell periphery with no persistent migration (*Ozawa et al., 2020*). In the cells spreading circularly, we noticed that the nuclei exhibit rotational motion in a clockwise direction when viewed from the dorsal (apical) side (*Figure 1B*). There was no cell that exhibited rotational motion in a counterclockwise direction. 24% of the cells exhibited migratory behavior at the start of our live imaging, and it took a while for the cells to spread circularly without persistent migration. The cells exhibiting migratory behavior were excluded from the analysis.

The speed of nuclear rotation was about 50 degrees/hr on average (*Figure 1C*). We measured the rotational speed by tracking unique points of the nuclear texture (*Figure 1A*). The texture of the cytoplasm around the nucleus also showed a rotating motion, which indicates that the cytoplasm circulates in the same direction (*Figure 1A*). Furthermore, we found that microbeads attached to the dorsal surface rotate (*Video 2*), confirming that the dorsal membrane also rotates. The rotating motion of cell nuclei persists for more than 8 hr until cell division occurs. After the cell division, cells form two-cell colonies, and then the nuclear rotation resumes. In this work, we focus on the rotating motion in singly isolated cells.

The rotational speed of the nucleus depended on the type of coating applied to the glass substrate, although the direction of rotation remained unaffected. When cells were cultured on fibronectin-coated glass, they exhibited the same clockwise nuclear rotation as observed on collagen-coated substrates, albeit at a slightly reduced speed (*Figure 1—figure supplement 1*). On poly-L-lysine-coated or uncoated glass, the rotation speed further decreased, with some cells exhibiting little rotation (*Figure 1—figure supplement 1*). Notably, while different coatings influenced the speed of nuclear rotation, they did not alter its direction.

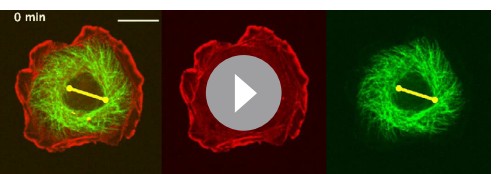

**Video 3.** Fluorescent images of Caco-2 expressing Lifeact-RFP and EMTB-3XGFP. Scale bar: 20 µm.
https://elifesciences.org/articles/102296/figures#video3

### F-actin and microtubules exhibit chiral patterns

We hypothesized that cytoskeletal molecules, such as F-actin and microtubules, are responsible for the circulating flow. To see the structure and dynamics of actin, we imaged live Caco-2 cells expressing Lifeact-RFP (*Video 1*). In the peripheral region of cells, actin bundles were tilted, forming a dextral chiral pattern (*Figure 1D*). Since each of these actin bundles appears to

associate with vinculin (*Figure 1—figure supplement 2*), a focal adhesion protein, at their termini, we refer to them as stress fibers (*Tojkander et al., 2012*). In more interior regions of the cell, actin filaments became thinner, losing their attachment to vinculin, and tended to adopt an orientation parallel to the cell periphery. Next, we observed microtubules by visualizing them with the GFP-tagged microtubule-binding domain of ensconsin (EMTB-3XGFP; *Miller and Bement, 2009*). Micro-tubules spread over the entire cytoplasmic region and exhibited a sinistral chiral pattern (*Figure 1D*, *Video 3*). A similar sinistral pattern can emerge when filaments extend radially from a center and the central region rotates clockwise, which is consistent with the direction of nuclear rotation. To summarize, actin bundles in the cell peripheral region and microtubules in the cytoplasm showed chiral patterns.

## Chiral rotation requires actomyosin activity, independent of chiral assembly of stress fibers

To investigate whether the cytoskeletons with chiral patterns drive the circulating flow, we performed live imaging of Caco-2 cells expressing Lifeact-RFP with small-molecule inhibitors of cytoskeletal structures. When cells were treated with the actin polymerization inhibitor latrunculin A or F-actin stabilizer jasplakinolide, the shape of the cell periphery became rough and nuclear rotation stopped (*Figure 2A*, *Figure 2—video 1*, and *Figure 2—video 2*, respectively), indicating that F-actin is necessary for the nuclear and cytoplasmic rotation. In contrast, disruption of microtubules by nocodazole did not affect the nuclear rotation (*Figure 2B, C*, *Figure 2—figure supplement 1*, and *Figure 2—video 3*), which indicates that microtubules are not involved in the rotating motion.

To reveal which activities of actin are involved in the chiral rotating motion, we first investigated the role of Arp2/3-driven actin polymerization on the rotating motion, since a previous report has shown that it is involved in the chiral behavior of HFF (*Tee et al., 2015*). When Caco-2 cells were treated with the Arp2/3 complex inhibitor CK666 (*Nolen et al., 2009*), lamellipodia at the cell periphery tended to shrink (*Figure 2A*, *Figure 2—video 4*), but the nuclear rotation was maintained (*Figure 2B, C*), indicating that the Arp2/3 complex was dispensable for the rotating motion, in contrast to HFF where the Arp2/3 complex plays a role in cell chirality formation (*Tee et al., 2015*).

Next, we focused on the potential role of formin, a regulator of actin polymerization, as previous studies showed that this protein is involved in inducing the chirality of some cell types (*Tee et al., 2015*; *Davison et al., 2016*; *Kuroda et al., 2016*; *Abe and Kuroda, 2019*; *Middelkoop et al., 2021*; *Tee et al., 2023*). To this end, we tested the effect of SMIFH2, a formin inhibitor (*Rizvi et al., 2009*), on the chiral pattern of Caco-2 cells. When cells were treated with this inhibitor, chiral stress fibers mostly disappeared in their peripheral regions, but instead, another pattern of F-actin appeared (*Figure 2A* and *Figure 2—video 5*). To investigate the distribution of F-actin more closely, we performed phalloidin staining and imaged cells in 3D (*Figure 3*). *Figure 3B* shows that a subset of actin bundles became oriented in a radial direction, unlike the chiral pattern of stress fibers originally observed in the control cells (*Figure 3A*). Furthermore, another population of F-actin was organized into a dense network or cluster with a concentric pattern (*Figure 3B*). Intriguingly, in spite of these drastic changes in the spatial organization of F-actin, and the disappearance of the peripheral chiral stress fibers, the rotating motion was maintained (*Figure 2B, C*). Furthermore, as shown in *Figure 2D*, we noticed that, while the nuclear rotation speeds of control and SMIFH2-treated cells were comparable in the first 5 hr of the observation window, the nucleus of the SMIFH2-treated cells rotated significantly faster than control cells, on average, in the second 5 hr: the rotating speed of control cells slightly decreased over time, while SMIFH2-treated cells maintained or even slightly accelerated the rotating speed (*Figure 2C, D*).

These observations with the formin inhibitor suggested that formins might not be essential for nuclear rotation. To further evaluate this idea, we knocked down formin expression using siRNA. Among the major mammalian formin family members (DIAPHs and DAAMs), DIAPH2 and DAAM1 were identified as being highly expressed in Caco-2 cells by RNA-sequencing (RNA-seq) analysis (*Figure 2—figure supplement 2A*). However, knockdown of either DIAPH2 or DAAM1 did not disrupt nuclear rotation (*Figure 2—figure supplement 2A–D*, *Figure 2—videos 6 and 7*). These findings suggest that formins are not essential for the chiral nuclear rotation in Caco-2 cells, in contrast to previous reports (*Tee et al., 2015*; *Davison et al., 2016*; *Kuroda et al., 2016*; *Abe and Kuroda, 2019*; *Middelkoop et al., 2021*; *Tee et al., 2023*).

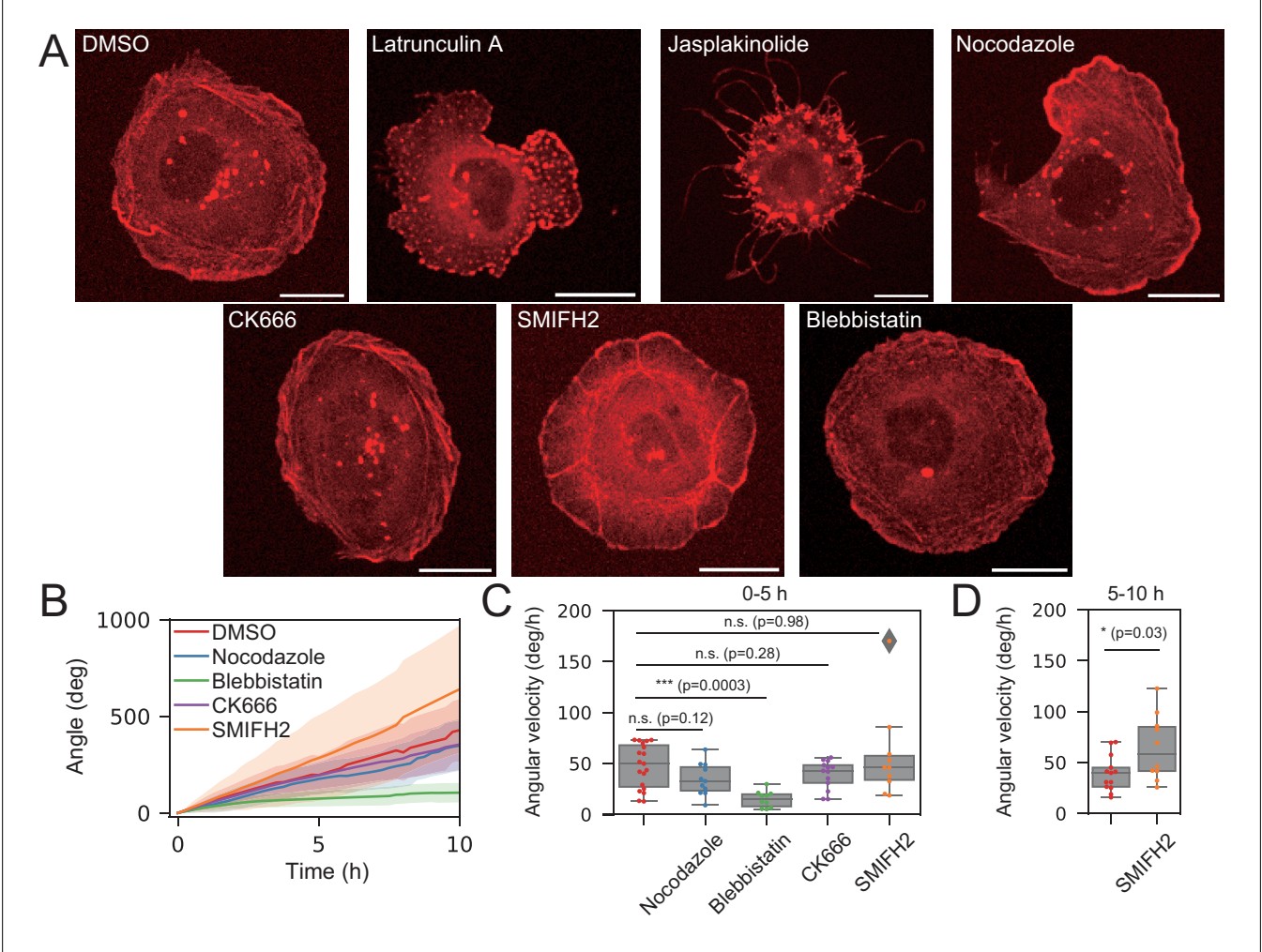

**Figure 2.** Identification of molecular mechanisms responsible for the chiral rotation. Roles of F-actin, microtubule, Arp2/3, formin-mediated actin polymerization, and Myosin II activity were investigated. Cells were treated with DMSO (0.2%, control), actin polymerization inhibitor latrunculin A (2 μM), actin depolymerization inhibitor Jasplakinolide (10 nM), microtubule inhibitor nocodazole (50 μM), Arp2/3 inhibitor CK666 (200 μM), formin inhibitor SMIFH2 (40 μM), or Myosin II inhibitor blebbistatin (1 μM). (**A**) Snapshot images from the live image of actin dynamics in cells expressing Lifeact-RFP. Scale bar: 20 μm. (**B**) The cumulative angle of nuclear rotation averaged over different cells plotted against time for different conditions: DMSO ($n = 19$), nocodazole ($n = 11$), blebbistatin ($n = 10$), CK666 ($n = 13$), and SMIFH2 ($n = 10$). The standard deviation is represented by shaded regions. (**C**) Angular velocity of cells under different conditions averaged over the first 5 hr of the time-evolution plot in (**B**). (**D**) Angular velocity of control and SMIFH2-treated cells averaged over the last 5 hr of the time-evolution plot in (**B**): DMSO ($n = 13$) and SMIFH2 ($n = 10$). p values were calculated using the Mann–Whitney $U$ test ($*p < 0.05, **p < 0.01, ***p < 0.001$). Here, positive angle values indicate clockwise rotation.

The online version of this article includes the following video, source data, and figure supplement(s) for figure 2:

**Figure supplement 1.** Microtubule inhibition by nocodazole.

**Figure supplement 2.** Effect of formin depletion on the nuclear rotation.

**Figure supplement 2—source data 1.** PDF file containing original western blots for *Figure 2—figure supplement 2B, C*, indicating the relevant bands and treatments.

**Figure supplement 2—source data 2.** Original files for western blot analysis displayed in *Figure 2—figure supplement 2B, C*.

**Figure supplement 3.** Effect of Myosin II depletion on the nuclear rotation.

**Figure supplement 3—source data 1.** PDF file containing original western blots for *Figure 2—figure supplement 3B*, indicating the relevant bands and treatments.

**Figure supplement 3—source data 2.** Original files for western blot analysis displayed in *Figure 2—figure supplement 3B*.

**Figure supplement 4.** Effect of vinculin depletion on the nuclear rotation.

**Figure supplement 4—source data 1.** PDF file containing original western blots for *Figure 2—figure supplement 4B*, indicating the relevant bands

*Figure 2 continued on next page*

*Figure 2 continued*

and treatments.

**Figure supplement 4—source data 2.** Original files for western blot analysis displayed in *Figure 2—figure supplement 4B*.

**Figure 2—video 1.** Differential interference contrast (DIC) and fluorescent images of Caco-2 expressing Lifeact-RFP treated with latrunculin A.
https://elifesciences.org/articles/102296/figures#fig2video1

**Figure 2—video 2.** Differential interference contrast (DIC) and fluorescent images of Caco-2 expressing Lifeact-RFP treated with jasplakinolide.
https://elifesciences.org/articles/102296/figures#fig2video2

**Figure 2—video 3.** Differential interference contrast (DIC) and fluorescent images of Caco-2 expressing Lifeact-RFP treated with nocodazole.
https://elifesciences.org/articles/102296/figures#fig2video3

**Figure 2—video 4.** Differential interference contrast (DIC) and fluorescent images of Caco-2 expressing Lifeact-RFP treated with CK666.
https://elifesciences.org/articles/102296/figures#fig2video4

**Figure 2—video 5.** Differential interference contrast (DIC) and fluorescent images of Caco-2 expressing Lifeact-RFP treated with SMIFH2.
https://elifesciences.org/articles/102296/figures#fig2video5

**Figure 2—video 6.** Differential interference contrast (DIC) and fluorescent images of Caco-2 expressing Lifeact-RFP treated with siRNA for DIAPH2.
https://elifesciences.org/articles/102296/figures#fig2video6

**Figure 2—video 7.** Differential interference contrast (DIC) and fluorescent images of Caco-2 expressing Lifeact-RFP treated with siRNA for DAAM1.
https://elifesciences.org/articles/102296/figures#fig2video7

**Figure 2—video 8.** Differential interference contrast (DIC) and fluorescent images of Caco-2 expressing Lifeact-RFP treated with blebbistatin.
https://elifesciences.org/articles/102296/figures#fig2video8

**Figure 2—video 9.** Differential interference contrast (DIC) and fluorescent images of Caco-2 expressing Lifeact-RFP treated with siRNA for Myosins IIA and B.
https://elifesciences.org/articles/102296/figures#fig2video9

**Figure 2—video 10.** Differential interference contrast (DIC) and fluorescent images of Caco-2 expressing Lifeact-RFP treated with siRNA for vinculin.
https://elifesciences.org/articles/102296/figures#fig2video10

In DIAPH2 knockdown cells, the distribution of actin fibers was similar to that in control cells (*Figure 2—figure supplement 2E, E′* and *Figure 2—video 6*). By contrast, a subset of DAAM1-knockdown cells displayed a detached end of the actin bundle, resembling the phenotype observed in SMIFH2-treated cells (*Figure 2—figure supplement 2F, F′* and *Figure 2—video 7*). Although the precise mechanism by which SMIFH2 induces F-actin reorganization while promoting nuclear rotation remains unclear, particularly given its lack of strict specificity for formins (*Nishimura et al., 2021b*), we employed this inhibitor as a tool to further investigate the mechanism underlying chiral rotational motion.

Previous studies have shown that the chirality in various cell types depends on the activity of Myosin II, proposing that Myosin II with F-actin generates chiral torque on a molecular scale (*Naganathan et al., 2014*; *Fürthauer et al., 2012*; *Fürthauer et al., 2013*; *Tjhung et al., 2017*). Therefore, we next investigated the role of Myosin II in the chiral rotation by treating Caco-2 cells with a Myosin II inhibitor, blebbistatin. Under this condition, the chiral pattern of peripheral F-actin became less prominent (*Figures 2A and 3C* and *Figure 2—video 8*), while the nuclear rotation was mostly suppressed (*Figure 2B, C*). To confirm the role of Myosin II, we depleted Myosin IIA and/or Myosin IIB heavy chains in Caco-2 cells using siRNAs (*Figure 2—figure supplement 3B–D*). Their depletion resulted in a significant reduction in the nuclear rotation (*Figure 2—figure supplement 3A*, *Figure 2—video 9*). These results suggest that the activity of Myosin II is required for the chiral rotational motion and also for the formation of a chiral pattern of stress fibers. In summary, both the activities of F-actin and Myosin II are important for nuclear and cytoplasmic rotation.

Our results using SMIFH2 showed that nuclear rotation persists even when the chiral stress fibers are lost, which suggests that these structures are not essential for driving rotational motion. To further test this possibility, we disrupted stress fibers by knocking down vinculin using siRNA. Upon vinculin depletion, the chiral arrangement of peripheral actin bundles was lost. Nevertheless, the nucleus continued to rotate clockwise comparable to that observed in control cells (*Figure 2—figure supplement 4* and *Figure 2—video 10*). These results indicate that Myosin II mediates chiral nuclear rotation through subcellular structures other than stress fibers.

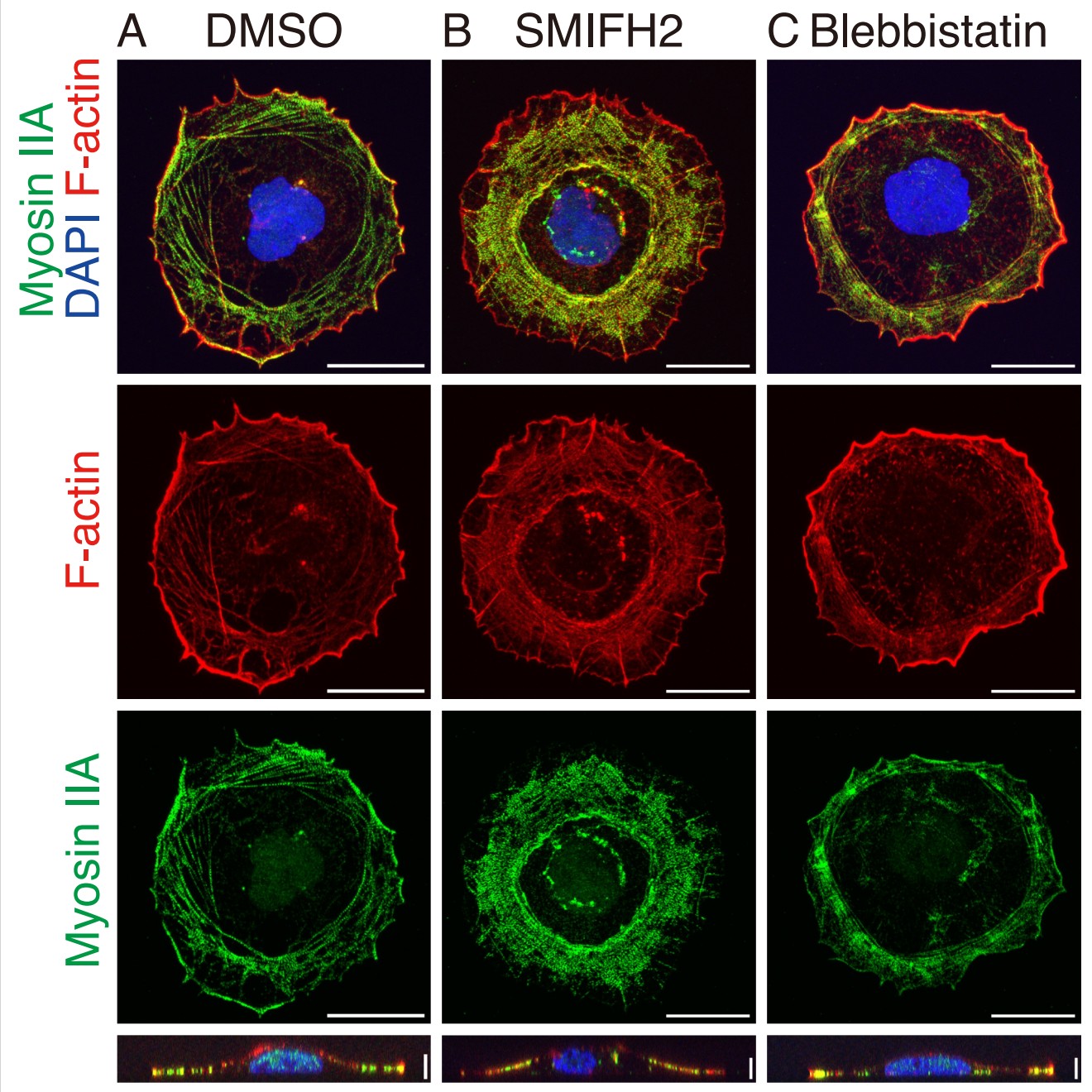

**Figure 3.** Organization of F-actin and Myosin IIA. (**A**) Control cells treated with DMSO show a chiral tilted pattern of F-actin and Myosin II visualized by phalloidin and immunofluorescence with an antibody against Myosin IIA, respectively. (**B**) SMIFH2 (40 μM) treated cells show a concentric pattern of F-actin and Myosin II. (**C**) The chiral tilted pattern of F-actin and Myosin II is suppressed in cells treated with blebbistatin (1 μM). The bottom panel shows vertical cross-sections. Scale bars: 20 μm (horizontal) and 5 μm (vertical).

## Super-resolution 3D imaging of actin and Myosin II

To further investigate the roles of F-actin and Myosin II in the chiral rotation, we analyzed their distribution and dynamics in more detail, using both control and SMIFH2-treated cells. For this purpose, we employed a 3D super-resolution imaging technique known as expansion microscopy (ExM).

We first examined the distribution of F-actin stained with phalloidin under control conditions (*Figure 4A, A′*). In the peripheral region of the cell, stress fibers localized on the ventral side exhibited a dextral swirling pattern (yellow in the left panel and bold lines in the right panel of *Figure 4A*; red in *Figure 4A′*; dark red lines in *Figure 5K*, top). These peripheral stress fibers (red in *Figure 4A′*) likely

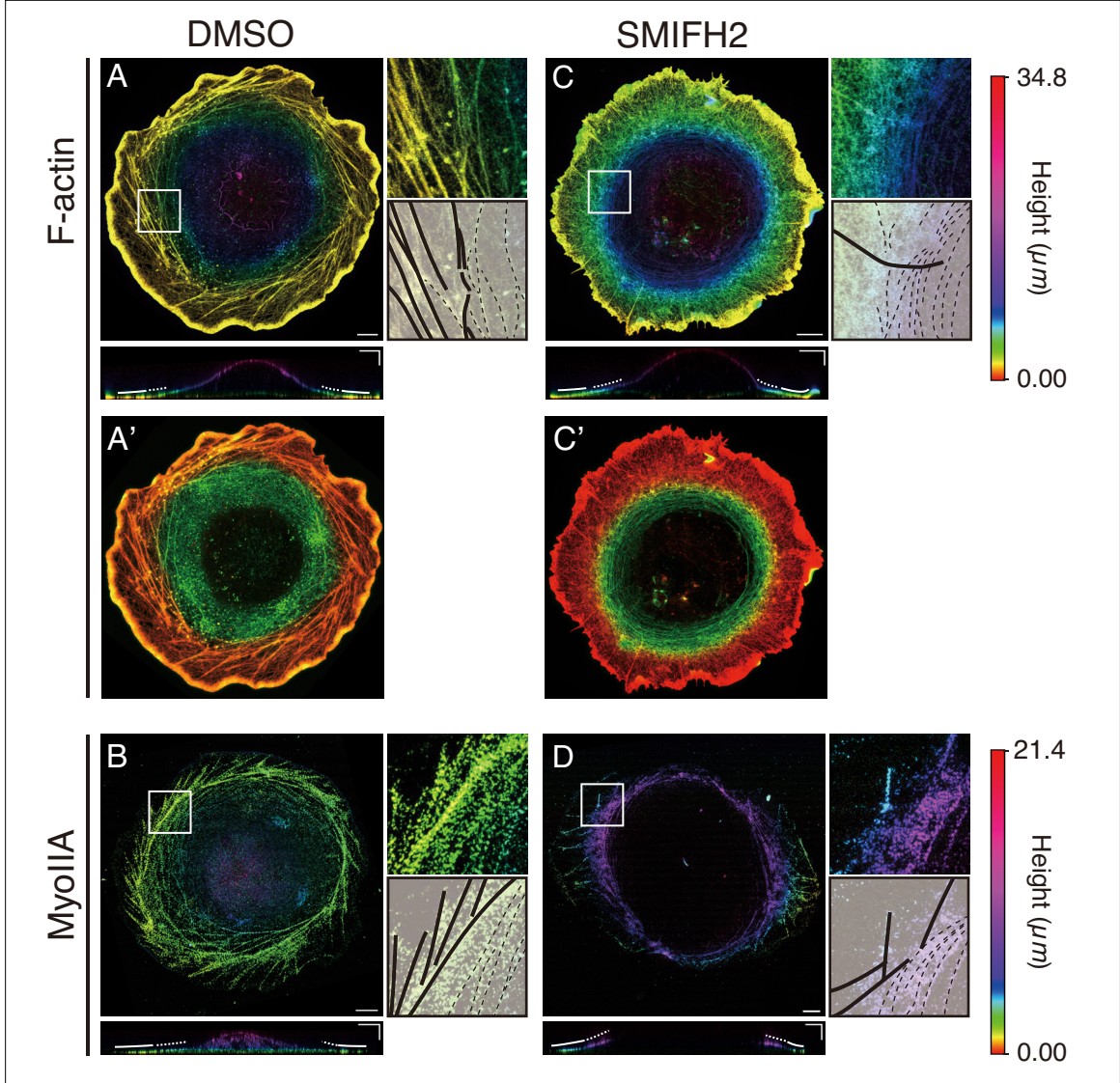

**Figure 4.** Expansion microscopy (ExM) imaging of F-actin and Myosin IIA. Maximum intensity projection (MIP) images of F-actin (**A, C**) and Myosin IIA (**B, D**) in DMSO (**A, B**) and SMIFH2 (**C, D**) treated cells. The color indicates the height along the *z*-axis, where the height was measured after the samples were swollen (color bar, right). Magnified views of the white boxes are shown in the right top panels, and corresponding outlines of F-actin are shown in the right bottom panels, where the bold and dotted lines indicate stress fibers and dorsal actomyosin fibers, respectively. The vertical cross-sections (*xz*) are shown in the bottom panels, where the bold and dotted lines indicate the peripheral and dorsal inner regions, respectively. Scale bars: 20 μm (horizontal) and 10 μm (vertical). (**A′, C′**) Composite F-actin images of the ventral (red) and dorsal (green) sides. (**A′**) In the DMSO-treated cell, the thickness of the ventral and dorsal sides is 2.7 and 6.5 μm, respectively. (**C′**) In the SMIFH2-treated cells, the thickness of the ventral and dorsal sides is 4.2 and 5.7 μm, respectively.

The online version of this article includes the following figure supplement(s) for figure 4:

**Figure supplement 1.** Expansion microscopy (ExM) imaging of actin filaments and Myosin II in cells treated with blebbistatin.

correspond to the actin bundles associated with focal adhesions, which were shown in *Figure 1—figure supplement 2*. Adjacent to the inner edge of the peripheral cytoplasmic zone having stress fiber bundles, we detected another population of actin filaments. These filaments, which looked thinner than stress fibers, were oriented parallel to the cell periphery and lacked obvious chirality, seemingly associated with the dorsal cell membranes (green in the left panel and dotted lines in the right panel of *Figure 4A*; light blue line in *Figure 5K*, top; green in *Figure 4A′*). These dorsal actin filaments (green in *Figure 4A′*) likely correspond to those not anchored to the focal adhesions, which were detectable in the image of *Figure 1—figure supplement 2*. They also appeared not to associate

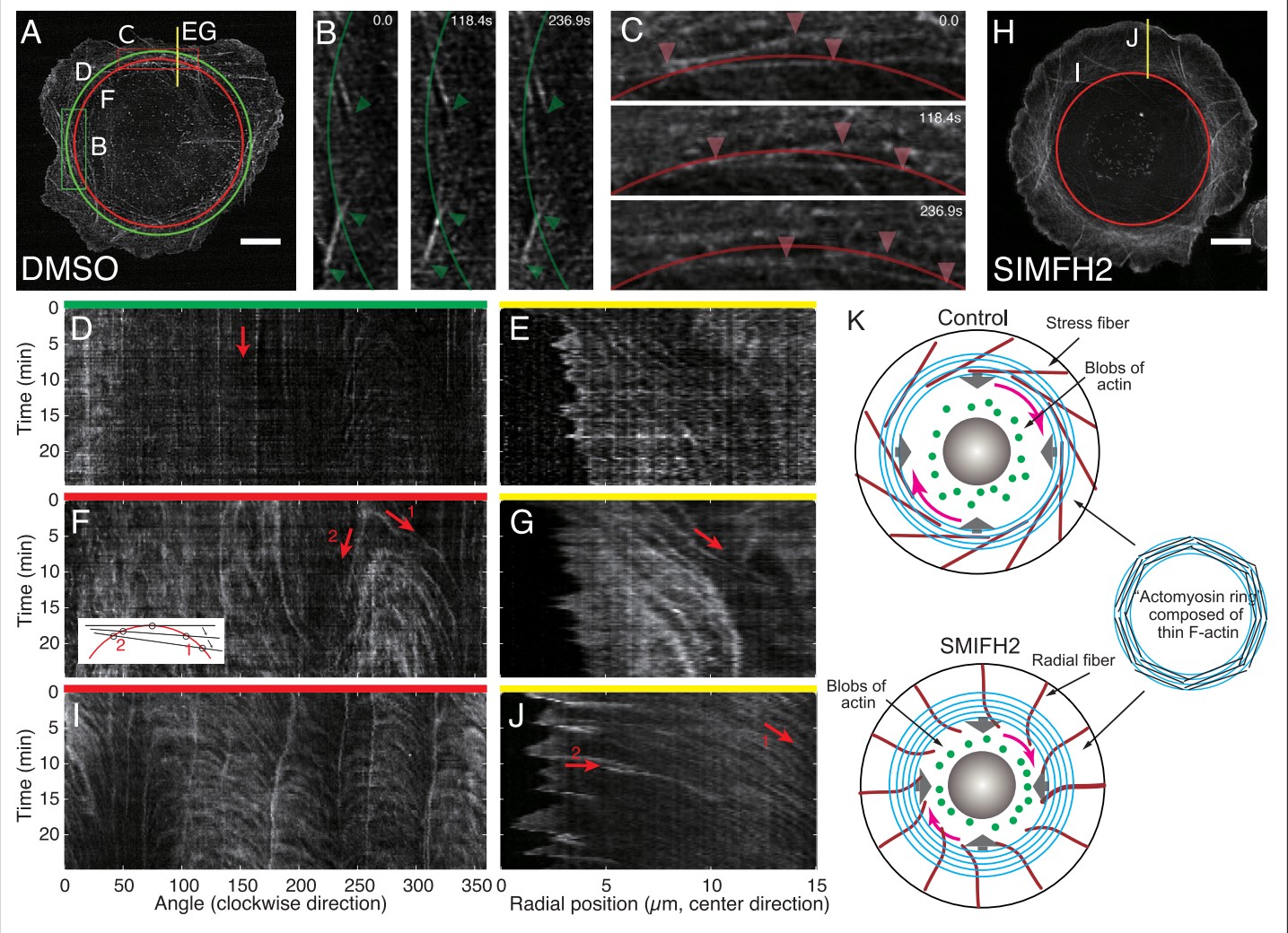

**Figure 5.** Dynamics of F-actin in Caco-2 cells live-imaged by lattice light-sheet microscopy (LLSM). (**A**) Maximum intensity projection (MIP) image of Caco-2 treated with DMSO. (**B**) Snapshot images in the green rectangle in (**A**), obtained from the $z$-slice at $z = 0$ is defined as the plane closest to the substrate. The green line is the same as the green circle in (**A**). Arrowheads indicate the position of filaments. (**C**) Snapshot images in the red rectangle in (**A**) obtained from the $z$-slice at $z = 0.5\,\mu$m. The red line is the same as the red circle in (**A**). Arrowheads indicate the position of filaments. (**D**) Kymograph along the green circle in (**A**), obtained from a slice at $z = 0$. (**E**) Kymograph along the yellow line in (**A**), obtained from the $z$-slice at $z = 0$. (**F**) Kymograph along the red circle in (**A**), obtained from the $z$-slice at $z = 0.5\,\mu$m. inset: schematic diagram of F-actin (black lines) passing through the circle. (**G**) Kymograph along the yellow line in (**A**), obtained from the $z$-slice $z = 0.5\,\mu$m. (**H**) MIP image of Caco-2 treated with SMIFH2 ($40\,\mu$m). (**I**) Kymograph along the red circle in (**H**), obtained from the MIP image. (**J**) Kymograph along the yellow line in (**H**), obtained from the MIP image. (**K**) Schematic diagram of F-actin structure in control and SMIFH2 treated cells. Stress fibers (dark red) were immobile, while the dorsal actin fibers formed an 'actomyosin ring' (light blue), moved in centripetal and clockwise directions. Scale bar: 10 µm.

with any other F-actin populations. Next, we examined the distribution of Myosin IIA using antibodies (*Figure 4B*). The distribution of Myosin IIA is similar to that of F-actin, except for its striped pattern. Since we could not assess the colocalization of F-actin and Myosin IIA filaments in the same cells in ExM for technical reasons, we investigated their localization using conventional confocal microscopy (*Figure 3A* and *Video 4*). The confocal microscopy images indicate that F-actin and Myosin IIA generally colocalize with one another in these specimens.

We next examined the distribution of F-actin and Myosin II in the cells treated with SMIFH2 (*Figure 4C, C', D*). As observed by conventional confocal microscopy, the chirally tilted actin stress fibers disappeared at the peripheral region, and instead, F-actin bundles extended radially from the cell edge toward its center (bold line in *Figure 4C*, right; dark red lines in *Figure 5K*, bottom). In the interior region, actin filaments were organized into a dense network with a concentric pattern,

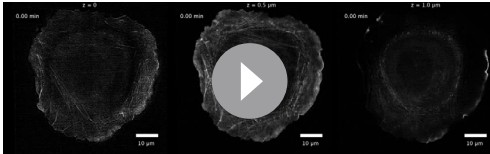

**Video 4.** Fluorescent images of Caco-2 expressing Lifeact-RFP treated and MRLC-EGFP. Scale bar: 20 μm. https://elifesciences.org/articles/102296/figures#video4

**Video 5.** Fluorescent images of Caco-2 expressing Lifeact-mEmelard taken by lattice light-sheet microscopy (LLSM) at three different z-planes (z = 0, 0.5, and 1.0 μm).

https://elifesciences.org/articles/102296/figures#video5

which was distributed at the dorsal side of cells (green in the left and dotted lines in the right panels of *Figure 4C*; light blue line in *Figure 5K*, bottom; green in *Figure 4C'*). Myosin IIA exhibited a similar reorganization as seen in F-actin (*Figure 4D*). As observed in control cells, confocal microscopy showed that F-actin and Myosin IIA also colocalize in the SMIFH2-treated cells, particularly in the concentric actin clusters (*Figure 3B*). Thus, ExM analysis revealed more detailed features of actomyosin distribution, particularly detecting its concentric orientation, located more inside the cell than the peripheral stress fibers.

Additionally, we examined cells treated with blebbistatin by ExM, confirming the results obtained by live imaging and conventional immunostaining (*Figures 2A and 3C*). The chiral orientation of stress fibers was greatly reduced in the peripheral region after this treatment (*Figure 4—figure supplement 1*). Furthermore, the dorsally located concentric actin filaments became undetectable in these specimens. These results are consistent with the idea that Myosin II plays a complex role, including in the organization of dorsal actin filaments.

## F-actin ring circulates along the dorsal membrane

To gain further insights into the role of the actomyosin system in the mechanism of intracellular rotation, we examined how F-actin behaves during the rotational process. To this end, we performed live imaging of Caco-2 cells expressing Lifeact-mEmerald, using lattice light-sheet microscopy (LLSM). Since LLSM has a higher spatial resolution, particularly in the z-direction, compared to conventional confocal microscopy, we could identify the dynamics of F-actin in 3D more precisely. We found that stress fibers at the ventral side were almost immobile (*Figure 5B*), while the actin fibers at the dorsal side moved clockwise as indicated in snapshot images (*Figure 5C*). Such spatiotemporal dynamics can be systematically seen in the kymographs along different lines, drawn in *Figure 5A*, at different heights $z$ in control cells. In *Figure 5D, E*, the kymographs along the green circle and the yellow line, which were analyzed at the ventral side, indicate that F-actin bundles in the peripheral region with the chiral tilted pattern are almost immobile (arrow in *Figure 5D*, *Video 5*). *Figure 5F, G* shows the kymographs along the red circle and the yellow line (drawn in *Figure 5A*), respectively, at the height where the dorsal cell membrane exists. Rightward descending lines in the kymograph along the red circle indicate that the filaments move in a clockwise direction (*Figure 5F*, arrow 1). There are also leftward descending lines that appear at the same time as the rightward descending lines appear but with different steepness (*Figure 5F*, arrow 2). These pairs of lines indicate that the filaments are moving clockwise as well as centripetally (*Figure 5F*, inset). Furthermore, the kymograph along the yellow line (*Figure 5A*) also confirms that the filaments are moving centripetally (arrow in *Figure 5G*). To

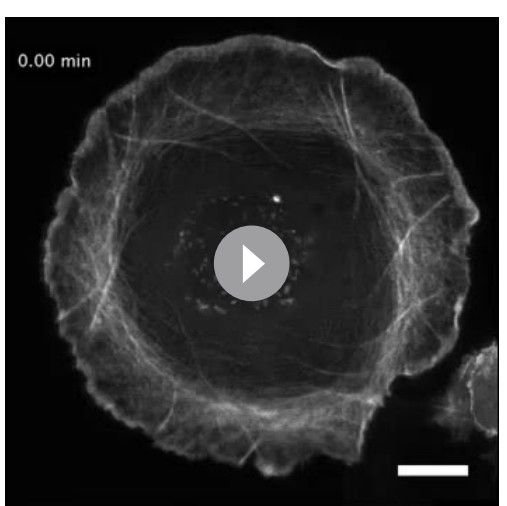

**Video 6.** MIP fluorescent images of Caco-2 expressing Lifeact-mEmelard treated with SMIFH2 taken by lattice light-sheet microscopy (LLSM). Scale bar: 10 μm.
https://elifesciences.org/articles/102296/figures#video6

summarize, along the dorsal cell membrane, the concentric actin filaments move clockwise while also moving in centripetally (see also Video F-actin ring circulates along the dorsal membrane): we hereafter call this concentric structure 'the actomyosin ring' (*Figure 5K*).

We also examined the dynamics of actin fibers in cells treated with SMIFH2, using live image data obtained by LLSM (*Figure 5H–J*). In *Figure 5I, J*, the kymographs along the red circle and yellow line (drawn in *Figure 5H*), respectively, indicate that the actomyosin filaments organizing the ring move clockwise (*Figure 5I*), and simultaneously flow centripetally (arrow 1 in *Figure 5J*), similar to the control condition. Additionally, we found that in contrast to the immobile stress fibers with a chiral pattern in control cells (dark red lines in *Figure 5K*, top), F-actin bundles radially extending from the cell periphery in SMIFH2-treated cells appeared to move passively in a clockwise direction at their proximal ends, although they seem to keep the anchorage of the distal ends to the cell edge (arrow 2 in *Figure 5J*; *Video 6* and dark red lines in *Figure 5K*, bottom), implying that these radial F-actin bundles do not play active roles in the chiral motion of the actin ring. Thus, the concentric ring of

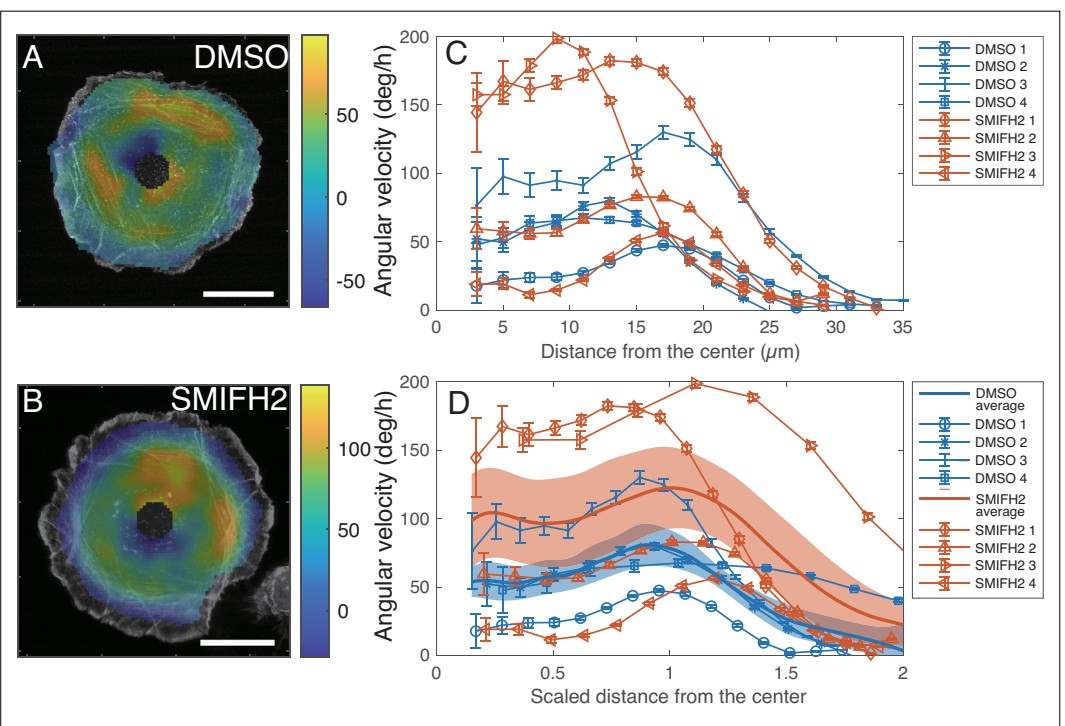

**Figure 6.** Angular velocity obtained by the particle image velocimetry (PIV) analysis. Spatial profile of angular velocity (color code) obtained from the time average of the PIV vector field in a control cell (**A**: DMSO) or in a cell treated with SMIFH2 (**B**) superimposed on a snapshot F-actin image. Scale bar: 20 μm. (**C**) Average angular velocity as a function of the distance from the center. (**D**) Average angular velocity as a function of a distance scaled by the inner radius of the actomyosin ring of individual cells. Here, positive angular velocity indicates clockwise rotation. Sample averages for two conditions are indicated by the solid lines. Error bars and shaded areas represent standard errors of the means (SEM).

The online version of this article includes the following video and figure supplement(s) for figure 6:

**Figure supplement 1.** Particle image velocimetry (PIV) analysis of cell treated with DMSO and SMIFH2.

**Figure 6—video 1.** Particle image velocimetry (PIV) analysis performed at each time step on the cell shown in Video F-actin ring circulates along the dorsal membrane.
https://elifesciences.org/articles/102296/figures#fig6video1

**Figure 6—video 2.** MIP fluorescent images of Caco-2 expressing Lifeact-mEmelard treated with SMIFH2 taken by LSM880, Zeiss.
https://elifesciences.org/articles/102296/figures#fig6video2

**Figure 6—video 3.** MIP fluorescent images of Caco-2 expressing Lifeact-mEmelard treated with SMIFH2 taken by lattice light-sheet microscopy (LLSM).
https://elifesciences.org/articles/102296/figures#fig6video3

flowing actomyosin filaments was detected also in the SMIFH2-treated cells but showing modified features (*Figure 5K*). Importantly, the ring developed more extensively after SMIFH2 treatment.

To see if the actomyosin ring is involved in driving the rotating flow, we estimated the spatial distribution of flow speed and orientation (velocity field) from the F-actin time-lapse images using particle image velocimetry (PIV) for both control and SMIFH2-treated cells (*Figure 6*, *Figure 6—figure supplement 1*, and *Figure 6—video 1*). In the cytoplasmic region between the actomyosin ring and the nucleus, we did not detect clear actin filaments, but only found blobs of actin (*Figure 5A, H*, and green dots in *Figure 5K*). These blobs also circulated clockwise. From the velocity field inside the cells inferred by PIV, we calculated the angular component of the velocity with respect to the cell center and then converted it into the angular velocity, that is change in the angle per unit time. The spatial profile of the angular velocity (*Figure 6A*, *Figure 6—figure supplement 1A–H*) indicates that it is higher in the region where concentric actin filaments are present, rather than the region where the actin blobs are present, indicating that the driving force could be present in the region of the actomyosin ring. The angular velocity averaged over the angular direction shows the peaks in the range from 10 to 20 μm (*Figure 6C*). Since the size of the actomyosin ring varies from cell to cell, we manually determined the region of the actomyosin ring, and then replotted the angular velocity against the distance scaled by the inner radius of the actomyosin ring (*Figure 6D*). We found that the peak positions of individual angular velocity profiles, as well as the peak positions of the angular velocity profiles averaged over samples, are located around the scaled distance of one, which suggests that the driving force is present in the region around the actomyosin ring.

We additionally observed another interesting phenomenon to support our idea. In an SMIFH2-treated cell that was imaged by a conventional fluorescence confocal microscope (LSM880, Zeiss), we, by chance, observed that fluorescent debris that seemed to attach to the actomyosin ring persistently circulated approximately three times as fast as the rotating speed of the nucleus: ~400 and ~140 degrees/hr for the debris and nucleus, respectively. (*Figure 6—video 2*, yellow and white lines, respectively). In the other cell observed by LLSM, we observed two fluorescent debris circulating in the area of the actomyosin ring, and in the cytoplasmic region between the actomyosin ring and the nucleus (*Figure 6—video 3*, yellow and red circles): the angular velocities of the circulating debris were ~250 and ~190 degrees/hr, respectively. Although we could not measure the rotating speed of the nucleus in the second cell because the nucleus was barely visible in the LLSM live image, the circulating speeds of the debris are more than two times faster than the typical nuclear angular velocity of SMIFH2-treated cells (*Figure 2B–D*). These observations support the notion that the actomyosin ring generates a driving force for rotating the nucleus and cytoplasm. Note that, since the angular velocity estimated from the motion of debris was faster than that obtained from the PIV analysis, our PIV analysis for F-actin dynamics may underestimate the flow velocity.

## A theoretical model of chiral cytoplasmic flow induced by the actomyosin ring

Our observations indicate the possibility that the actomyosin ring is a cell-scale structure that drives the chiral cytoplasmic flow. Since the actin filaments of the ring seemed not to have contact with the peripheral stress fibers in control cells, their rotational motion should be driven solely through its own mechanism, without relying on other structures, contrasted with the previous model that the contact between transverse fibers and radial fibers plays a role in establishing the cell chirality in HFF (*Tee et al., 2015*). Then, how can a concentric pattern without an obvious cell-scale chiral structure generate chiral circulating flow? We here theoretically address this question.

We employ a theoretical framework of active chiral fluid (*Fürthauer et al., 2012*; *Fürthauer et al., 2013*; *Naganathan et al., 2014*), which has been proposed to describe the fluid dynamics driven by active chiral components. We model the actomyosin ring as an active chiral fluid driven by two active elements: (1) a force dipole originating from the contraction force of actomyosin and (2) a torque dipole generated when a bipolar Myosin II filament rotates two antiparallel actin filaments to create a pair of counter-rotating vortex flows (*Figure 7A*). By representing the orientation of actomyosin fibers as orientational field $\mathbf{p}$ and the fluid velocity as $\mathbf{v}$, the hydrodynamic equation is described by the Stokes equation with the active contributions:

$$0 = -\nabla P + \eta \nabla^2 \mathbf{v} + \zeta^a \nabla \cdot \mathbf{pp} + \frac{1}{2}\zeta^c \nabla \times (\nabla \cdot \mathbf{pp}), \tag{1}$$

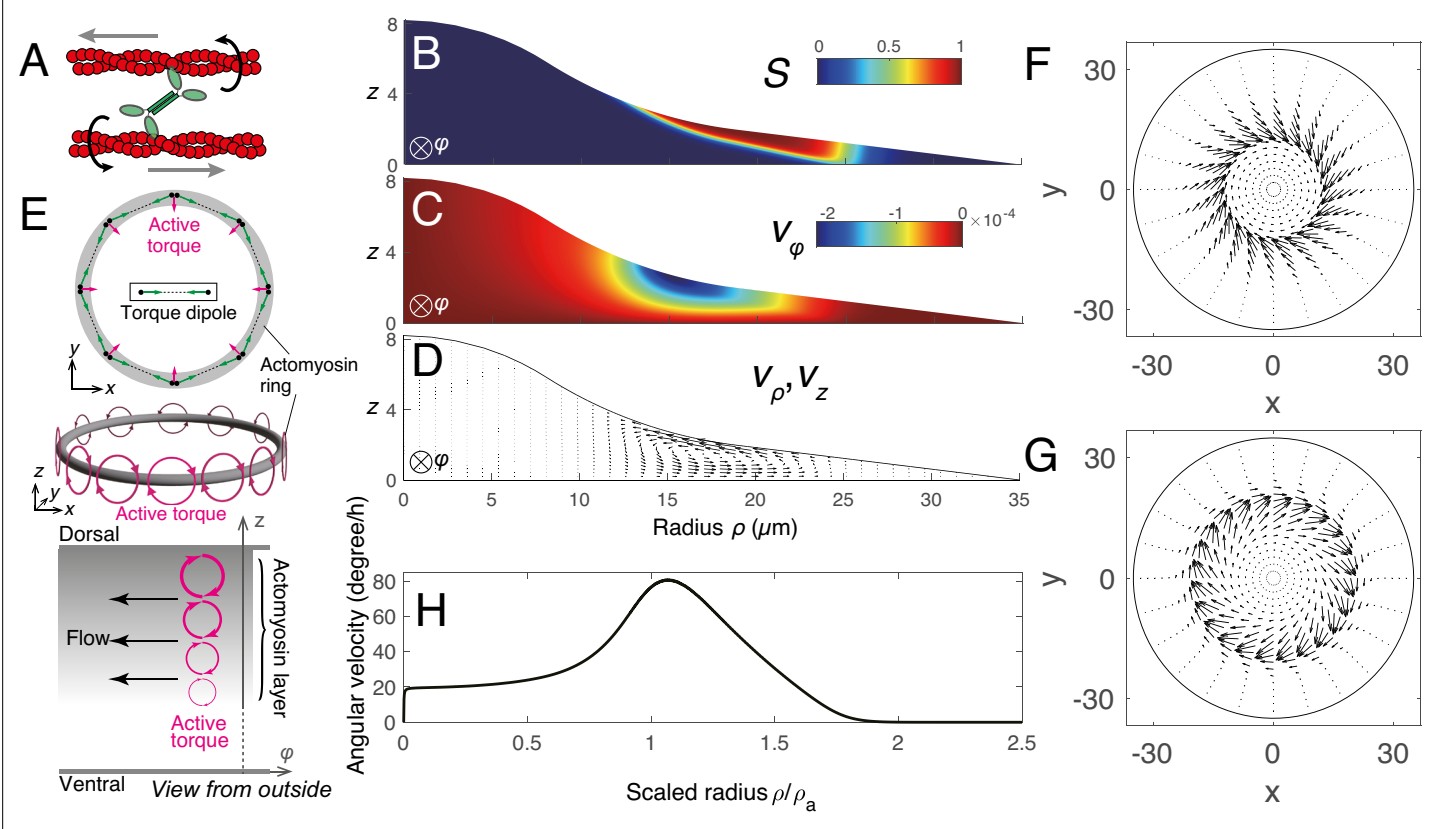

**Figure 7.** Theoretical model for the chiral cytoplasmic flow. (**A**) Actomyosin generates a force dipole and a torque dipole. (**B–D**) Numerical simulation of *Equation 1* assuming that the cell shape is axisymmetric around the cell center. (**B**) Actomyosin is distributed along the dorsal membrane with a concentric orientation. (**C**) Azimuthal velocity $v_\varphi$ showing negative values indicating that the flow is generated in a clockwise direction. (**D**) Velocity in the radial $\rho$- and $z$- directions, ($v_\rho$, $v_z$), indicated by vectors. Circulating flow is generated in the $\rho$-$z$ plane. (**E**) Top: A concentric orientational field on a ring generates active torque in the center direction (magenta arrow). Middle: Active torque (magenta clockwise arrow) generated by a concentric orientational field on a ring. Bottom: A side view of a cell from the outside toward the cell center. The concentration of actomyosin increases in $z$ (gray color), leading to a gradient of active torque (magenta clockwise arrow) in the $z$ direction, resulting in a rotational flow clockwise (black arrows). (**F**) Flow profile along the dorsal side showing an inward sinistral swirling pattern. (**G**) Flow profile at the ventral side showing an outward dextral swirling pattern. (**H**) Angular velocity averaged in the $z$ direction plotted along the radial direction, showing a peak at around $\rho/\rho_a = 1$. Here, $\rho_a$ is the leftmost position where $S \geq 0.8$.

The online version of this article includes the following figure supplement(s) for figure 7:

**Figure supplement 1.** Cell shape used in the numerical simulation.

**Figure supplement 2.** Comparison of radial and azimuthal velocities to determine the model parameters.

where $P$ is the pressure satisfying the incompressibility condition $\nabla \cdot \mathbf{v} = 0$ and $\eta$ is the fluid viscosity. Here, the terms with $\zeta^a$ and $\zeta^c$ are forces generated by the force dipoles (achiral) and torque dipoles (chiral), respectively. $\zeta^a$ and $\zeta^c$ represent the strength of the forces. The signs of $\zeta^a$ and $\zeta^c$ would be determined by the nature of force and torque generation at the molecular scale, independently of the cell-scale orientation of actomyosin. Considering that the actomyosin generates contractile force and right-handed torque as shown in *Figure 7A*; *Nishizaka et al., 1993*, the signs of the coefficients are $\zeta^a > 0$ and $\zeta^c > 0$. We hereafter assume $\zeta^a > 0$ and $\zeta^c > 0$ constant in space. We also assume the actomyosin filaments have a bipolar structure (*Figure 7A*) so that *Equation 1* is invariant under $\mathbf{p} \rightarrow -\mathbf{p}$. We, for convenience, represent the spatial variation of the density and order of the actomyosin by introducing an effective order parameter $S$ as $\mathbf{p} = S\mathbf{n}$ (see *Equation 11*), where $\mathbf{n}$ is a unit vector.

We first numerically solved *Equation 1*, assuming a cell with an axisymmetric geometry shown in *Figure 7—figure supplement 1*. Based on the experimental observation, we also assumed that actomyosin is distributed on the dorsal side, where the effective order parameter $S$ is set to be positive reflecting the density distribution of actomyosin (*Figure 7B*). *Figure 7C* shows the spatial profile of

the azimuthal velocity $v_\varphi$ in the vertical section of the cell. In the entire region, $v_\varphi$ is negative, indicating that the flow is generated in a clockwise direction viewed from above. Thus, the numerical result shows that the concentric pattern of actomyosin can generate chiral cytoplasmic flow, and the direction is clockwise, consistent with our experimental observation.

How can we understand the underlying mechanism behind the chiral cytoplasmic flow resulting from the concentric pattern of actomyosin? The active chiral term $\zeta^c \nabla \times \nabla \cdot \mathbf{pp}$ in *Equation 1* can be interpreted as follows: the rotation of the axial vector field $\zeta^c \nabla \cdot \mathbf{pp}$, which is an active torque induced by chiral torque dipole, generates a force to induce a flow. For a concentric orientational field on a ring domain, the active torque $\zeta^c \nabla \cdot \mathbf{pp}$ is generated, which is the clockwise direction when viewed from the outside toward the center of the ring, as shown in *Figure 7E* middle (magenta arrow). The actomyosin ring formed along the dorsal side in Caco-2 cells is regarded as a stack of concentric orientational fields on a ring. In the region where actomyosin is present ($S > 0$), the concentration of actomyosin naturally increases with $z$ which leads to an increase of the effective order parameter $S$ with respect to $z$ as indicated in *Figure 7B* (the gradient of red color in *Figure 7E*, bottom). For such an orientational field, the strength of the active torque increases as the height $z$ increases (magenta clockwise arrow in *Figure 7E*, bottom), forming a gradient of the active torque strength in the $z$ direction. Consequently, the gradient generates a force in a clockwise direction as viewed from above (black arrows in *Figure 7E* bottom, see also the right-hand side of *Equation 12*).

In the numerical simulation, we also investigated the spatial profile of $\rho$ and $z$ components $(v_\rho, v_z)$ of the velocity field. *Figure 7D* shows that inward flow to the cell center occurs on the dorsal side, while the flow in the opposite direction occurs on the ventral side, resulting in the circulating flow in the $\rho$ and $z$ plane (*Figure 7D*). This circulating flow is driven by the contractile force of actomyosin on the dorsal side. Due to the circulating flow in the $\rho$-$z$ plane and the azimuthal flow, swirling flows appear on both dorsal and ventral sides (*Figure 7F, G*, respectively). The sinistral swirling pattern at the dorsal side shown in *Figure 7F* is driven by both centripetal and clockwise flows, which were indeed observed by LLSM shown in *Figure 5*. Interestingly, the dextral swirling pattern of the flow on the ventral side is consistent with the chiral pattern of the stress fibers on the ventral side (*Figure 1D*). When the chiral flow was inhibited with blebbistatin, such a chiral pattern at the ventral side disappeared (*Figures 2A and 3C* and *Figure 4—figure supplement 1*), suggesting that the ventral chiral tilted patterns are indeed formed in a flow-dependent manner. This implies that the dextral chiral pattern of the stress fibers is self-organized through alignment with the fluid flow on the ventral side.

Furthermore, we analyzed the radial distribution of the angular velocity and found that it exhibits a peak around the inner edge of the actomyosin ring $\rho \sim \rho_a$ (*Figure 7H*) with the peak value of around 80 degree/h, consistent with the PIV analysis of the experimental data (*Figure 6C, D*). $\rho_a$ is the inner radius of the actomyosin ring. This result further supports the idea that the actomyosin ring drives the rotation of the nucleus.

## Depletion of dorsal actin and myosin coincides with the cessation of nuclear rotation

Our experimental observations and theoretical results suggest that the actomyosin ring located at the dorsal side of Caco-2 cells plays a key role in the rotating motion. To further investigate this, we tested whether depletion of actomyosin at the dorsal side affected rotational motion. A previous study showed that the activation of RhoA by Rho Activator II (CN03), a specific RhoA activator, resulted in a decrease in apical stress fibers and an increase in basal stress fibers in vascular smooth muscle cells (*Bade et al., 2017*). Based on this information, we aimed to decrease the ratio of the actomyosin at the dorsal to ventral sides in Caco-2 cells using Rho Activator II. Remarkably, we observed a substantial increase in the thickness and number of actomyosin bundles at the ventral side, particularly beneath the nucleus, in Caco-2 cells treated with Rho Activator II, while the dorsal actomyosin appeared to decrease significantly (*Figure 8A, C*). The ratio of dorsal to ventral actomyosin was reduced significantly in the cell treated with Rho Activator II compared with control cells (*Figure 8D, E*). We then performed live imaging of Caco-2 cells treated with Rho Activator II and found that the rotational motion of the nucleus ceased (*Figure 8—video 1*), supporting the idea that the dorsal actomyosin is crucial for driving the rotation. We also noticed that in cells treated with Rho Activator II, the bundle of stress fibers was rearranged into a chordal pattern (*Figure 8C*) and exhibited chiral motion (*Figure 8—video 1*), the mechanism of which remains to be understood. On the other hand,

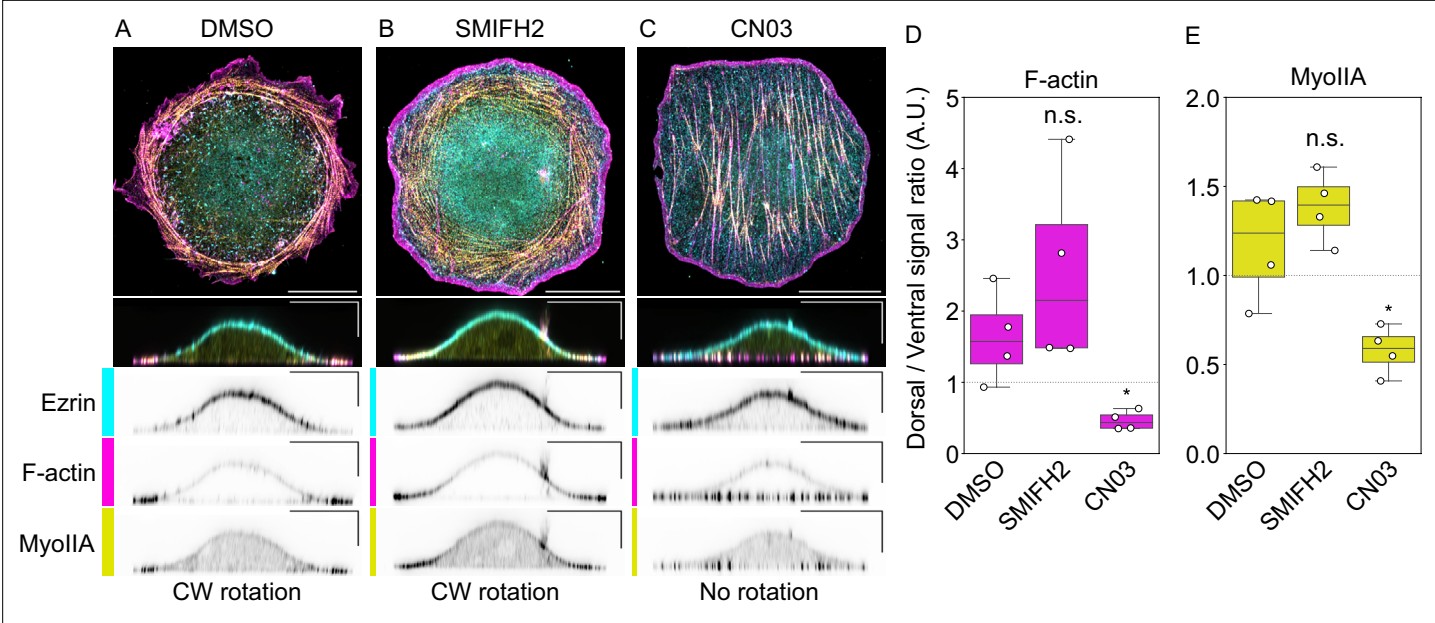

**Figure 8.** Depletion of dorsal actin and myosin by Rho Activator II stops the rotating motion of the nucleus. (**A**) Actin (magenta) and Myosin II (yellow) showing localization with the dorsal marker Ezrin (cyan) in the DMSO-treated cell. (**B**) SMIFH2-treated cell showing an increase in dorsal actomyosin and a decrease in ventral actomyosin. (**C**) Rho Activator II (CN03) treated cell showing a decrease in dorsal actomyosin and an increase in ventral actomyosin. Control (DMSO) and SMIFH2-treated cells showed clockwise (CW) rotation, while CN03-treated cells did show rotation. Scale bars: 20 μm (horizontal) and 5 μm (vertical). Ratio of dorsal F-actin (**D**) and MyoII (**E**) to the ventral ones. p values were calculated using the Mann–Whitney *U* test. ∗*p* < 0.05, n.s.: *p* ≧ 0.05.

The online version of this article includes the following video for figure 8:

**Figure 8—video 1.** Differential interference contrast (DIC) and fluorescent images of Caco-2 expressing Lifeact-RFP treated with CN03.
https://elifesciences.org/articles/102296/figures#fig8video1

in cells treated with SMIFH2, the dorsal actomyosin appeared to increase, while the ventral actomyosin decreased (*Figure 8A, B*). The ratio of dorsal to ventral actomyosin in cells treated with SMIFH2 tended to increase, although the difference was not statistically significant (*Figure 8D, E*). Taken together, our findings further support the idea that actomyosin at the dorsal side is crucial for driving rotation in Caco-2 cells.

## Discussion

In this study, we investigated the mechanism underlying cell-scale chiral dynamics, which is observed in Caco-2 epithelial cells when cultured as a single cell. We found that Caco-2 cells exhibited nuclear rotation and cytoplasmic circulation clockwise, and these movements require actin and Myosin II activities. High-resolution microscopy has revealed that the concentric actomyosin ring located on the dorsal side of the cells moves in a clockwise direction, leading us to hypothesize that this process may play a critical role in driving cytoplasmic flow. Previous studies using HFF proposed that radial actin fibers produce a force to drive the movement of the concentric actomyosin filaments (transverse fibers) through their connections (*Tee et al., 2015*). In the case of Caco-2 cells, however, we did not detect such radial fibers crossing the concentric actomyosin ring, and our observations suggested that the actomyosin ring by itself was moving in a clockwise direction through its own mechanism. To test this idea, we employed active chiral fluid theory (*Fürthauer et al., 2012*; *Fürthauer et al., 2013*; *Naganathan et al., 2014*), showing that the actomyosin localized under the dorsal membrane induces an active unidirectional fluid flow of the viscous cytoplasm and in turn nuclear rotation. Since the concentric pattern of actomyosin has no chirality at the cellular scale, our theory indicates that the nuclear rotation and cytoplasmic flow in Caco-2 cells is driven by the molecular-scale chiral mechanics of actomyosin rather than the cell-scale chiral orientation of actomyosin. It is also of note that we did not detect any visible cytoskeletal linkage between the nucleus and other cellular structures, another

potential machinery for driving nuclear rotation. The nuclear rotation may be induced directly by the cytoplasmic circulating flow mediated by the friction between the nuclear surface and the cytoplasm. Microtubules and the peripheral stress fibers also exhibited chiral distribution patterns, but we obtained no evidence that they are involved in the chiral motion of the nucleus and cytoplasm. It is, therefore, possible that their chiral distribution was generated as a result of the above-mentioned mechanism.

Our experiments using an inhibitor and RNAi-mediated depletion have revealed that Myosin II is involved in the chirality of Caco-2 cells, which is consistent with previous studies showing the involvement of Myosin II in the chiral behaviors of several types of cells (*Kumar et al., 2014*). Interestingly, while some of these studies concluded that formins are essential for breaking the chiral symmetry (*Tee et al., 2015*; *Davison et al., 2016*; *Kuroda et al., 2016*; *Abe and Kuroda, 2019*; *Middelkoop et al., 2021*; *Tee et al., 2023*), our results showed that the rotational speed in Caco-2 cells did not decrease but even slightly increased when treated with SMIFH2, a known inhibitor of formins, suggesting that formins are not required for Caco-2 cell chirality. Curiously, a previous study showed that SMIFH2 inhibits the centripetal movement of Myosin II filaments in HFF (*Nishimura et al., 2021b*), simultaneously demonstrating that this inhibitor also inhibits myosins, which apparently contradicts our observation. However, the same group also reported that SMIFH2 facilitated the centripetal movement of actin and myosin filaments when rat embryo fibroblasts were used (*Nishimura et al., 2021a*), similar to our present observations as shown in (*Figure 6—figure supplement 1J*). These reports suggest that how cells respond to SMIFH2 depends on their types. In the case of Caco-2 cells, it is likely that the observed effects of SMIFH2 on actomyosin dynamics were not attributed to its potential inhibition of Myosin II, although how this inhibitor induced the observed reorganization of actin and myosin filaments in Caco-2 cells remains unknown, as SMIFH2 seems to have multiple targets (*Nishimura et al., 2021a*). Our experiments to deplete DIAPH2 and DAAM1 also support the idea that formins are not essential for breaking the chiral symmetry, at least in the present cell system.

In Caco-2 cells treated with SMIFH2, the actomyosin ring became more visible than in the control cells (*Figures 3, 4, and 8*). Cells treated with SMIFH2 tended to show a trend of actin and myosin shifting from the ventral side to the dorsal side compared to control cells, although not statistically significant (*Figure 8D, E*). This might be due to a decrease in the formation of stress fibers on the ventral side, followed by a shift of free actin and Myosin II to the dorsal side, which may lead to an increase in the formation of actomyosin ring at the dorsal side. In contrast, treatment by Rho Activator II increased the stress fiber formation at the ventral side, which can induce a shift of actin and Myosin II to the ventral side, leading to a decrease in the formation of actomyosin ring at the dorsal side (*Figure 8D, E*). These considerations provide support for the model that dorsal actomyosin constitutes the driving force behind the rotational motion, as the rotation tended to increase or decrease in SMIFH2- or Rho Activator II-treated cells, respectively.

In a previous study (*Kumar et al., 2014*), an achiral active fluid model was proposed to explain the rotation of the nucleus driven by actomyosin. In the theory, the concentric orientational order of actomyosin becomes unstable due to the spontaneous chiral symmetry breaking induced by the contractility of actomyosin, and then a chiral orientational order emerges to drive a unidirectional fluid flow to rotate the nucleus. Since there is no intrinsic chirality in the model, either clockwise or counterclockwise rotation is selected with equal probability. In contrast, in our theoretical model, we considered the intrinsic chirality of the actomyosin gel in order to explain our experiments where the rotational direction is always clockwise. Although our model is consistent with our experimental observations, there is still a limitation. We assumed a concentric orientational order of actomyosin. However, it remains to be elucidated how the concentric order is formed (*Tarama and Shibata, 2022*; *Ni et al., 2022*), and how stable the structure is when the actomyosin has intrinsic chirality.

Several types of cells have been reported to exhibit chiral nuclear rotation. Zebrafish melanophores exhibit a counterclockwise nuclear rotation 'from basement view' (*Yamanaka and Kondo, 2015*), which is the same as our observation where the nucleus rotates in a clockwise direction viewed from the dorsal side. The study reported that actin plays a pivotal role in the chiral rotational motion. In the case of singly isolated MDCK cells embedded in a 3D culture, nuclei as well as whole cells exhibited rotations in either direction with a bias to the counterclockwise direction (*Chin et al., 2018*). Furthermore, a weak inhibition of actin polymerization reversed the bias in the rotational motion, and an actin-binding protein α-actinin-1 regulates the direction of chiral rotation (*Chin et al., 2018*), similar

to the case of HFF (*Tee et al., 2015*; *Tee et al., 2023*). In the case of C2C12 myoblasts, whether actin filaments were organized or disorganized was shown to correlate with chirally biased nuclear rotation (*Kwong et al., 2019*). In contrast to HFF (*Tee et al., 2015*; *Tee et al., 2023*), the direction of nuclear rotation of Caco-2 is opposite and the chirality of Caco-2 does not require the activity of Arp2/3 complex (*Figure 2*) nor physical interactions between the actin ring and other actin structures such as radial fibers. Thus, the mechanism of cell chirality formation seems to differ between cell types, such as HFF and Caco-2 cells. Whether there is a common principle behind the chiral nuclear rotation or whether there are multiple mechanisms remains to be clarified in future studies.

Beads attached externally to the dorsal membrane exhibited both chiral rotation and centripetal movement, as shown in *Video 2*. This behavior mirrors the movement of actin filaments at the dorsal side, suggesting that the dorsal membrane moves in concert with the underlying actomyosin. In contrast, stress fibers located just above the ventral membrane did not move as shown in the leftmost panel of *Video 5*, suggesting that the ventral membrane is likely immobile. These observations imply that the dorsal region of the cell undergoes twisting relative to the ventral region. Given the fluid-like properties of the membrane, such twisting is expected to be resolved over the time scale of the rotation of the dorsal side. Similar twisting behavior has been observed in zebrafish melanophores (*Yamanaka and Kondo, 2015*).

The observation that rotational speed changes with substrate coating (*Figure 1—figure supplement 1*) may be attributed to several factors, including redistribution of actomyosin that affects the amount of dorsal actomyosin, as exemplified by the effect of Rho Activator II, and substrate-dependent changes in the cell's internal physical environment, such as variations in effective viscosity at the ventral side.

It has been suggested that the left–right asymmetry at the tissue level originates from chirality at the cellular level. However, it remains unclear how cell chirality coordinates to induce left–right asymmetry in multicellular organisms. Our previous theoretical studies showed that a tissue-scale asymmetry, such as a spatial gradient in the strength of cell-scale torque generation, is necessary for the tissue-scale left–right asymmetry to arise from cell chirality (*Yamamoto et al., 2020*; *Sato et al., 2015b*; *Sato et al., 2015a*). Hence, it will be necessary to investigate how cell chirality and tissue-level asymmetry coordinate to understand left–right asymmetry in organs and the body. Caco-2 is an epithelial line that can form multicellular layers. We thus hope that investigating the coordination between cell chirality and tissue-level chirality using cells, such as Caco-2, which exhibit a clear individual chirality, is a promising approach to reveal the principles of chiral morphogenesis.

## Materials and methods

**Key resources table**

| Reagent type (species) or resource | Designation | Source or reference | Identifiers | Additional information |
|---|---|---|---|---|
| Cell line (*Homo sapiens*) | Caco-2 | ATCC | | *Ozawa et al., 2020* |
| Transfected construct (*Homo sapiens*) | Lifeact-RFP | *Ozawa et al., 2020* | | |
| Transfected construct (*Homo sapiens*) | EMTB-3XGFP | Addgene, *Miller and Bement, 2009* | Plasmid \# 26741 | |
| Transfected construct (*Homo sapiens*) | Lifeact-mEmerald | *Nakamura et al., 2012* | pLVSIN-EF1a-Lifeact-mEmerald-IRES-pur | |
| Sequence-based reagent | siRNA: Myosin IIA | Invitrogen | MYH9HSS106871 | |
| Sequence-based reagent | siRNA: Myosin IIB | Invitrogen | MYH10HSS106875 | |
| Sequence-based reagent | siRNA: DAAM1 | Invitrogen | DAAM1HSS177085 | |
| Sequence-based reagent | siRNA: DIAPH2 | Invitrogen | DIAPH2HSS102773 | |
| Sequence-based reagent | siRNA: Vinculin | Invitrogen | VCL s14764 | |
| Antibody | anti-Myosin IIA (rabbit monoclonal) | Sigma-Aldrich | M8064 | IF (1:1000) WB (1:1000) |

*Continued on next page*

*Continued*

| Reagent type (species) or resource | Designation | Source or reference | Identifiers | Additional information |
|---|---|---|---|---|
| Antibody | anti-Myosin IIB (rabbit monoclonal) | Cell Signaling Technology | 8824 | WB (1:1000) |
| Antibody | anti-GAPDH (mouse monoclonal) | Santa Cruz Biotechnology | sc-166574 | WB (1:1000) |
| Antibody | HRP-conjugated anti-rabbit IgG (goat monoclonal) | Invitrogen | T20926 | WB (1:1000–1:10,000) |
| Antibody | HRP-conjugated anti-mouse IgG (goat monoclonal) | Invitrogen | T20912 | WB (1:1000–1:10,000) |
| Antibody | anti-Vinculin (mouse monoclonal) | Sigma | V9131 | IF (1:200) WB (1:1000) |
| Antibody | anti-Ezrin (mouse monoclonal) | Abcam | ab4069 | IF (1:1000) |
| Antibody | anti-DAAM1 (mouse monoclonal) | Santa Cruz Biotechnology | sc-100942 | WB (1:1000) |
| Antibody | anti-Diaph2 (mouse monoclonal) | Santa Cruz Biotechnology | sc-55540 | WB (1:500) |
| Antibody | Alexa Fluor 488 anti-rabbit IgG (goat monoclonal) | Sigma-Aldrich | 11034 | IF (1:1000) |
| Antibody | Alexa Fluor 488 anti-mouse IgG (goat monoclonal) | Invitrogen | A11029 | IF (1:1000) |
| Antibody | Alexa Fluor 647 anti-mouse IgG (donkey monoclonal) | Sigma-Aldrich | AP192SA6 | IF (1:1000) |
| Antibody | anti-Alexa Fluor 488 (rabbit monoclonal) | abcam | ab150077 | IF (1:500) |
| Antibody | Alexa Fluor 488-conjugated anti-rabbit IgG (goat monoclonal) | *Park et al., 2020* | A11034 | IF (1:1000) |
| Chemical compound, drug | Latranculin A | Sigma | L5163-100UG | |
| Chemical compound, drug | Jasplakinolide | Toronto Research Chemicals Inc | J210700 | |
| Chemical compound, drug | Nocodazole | Sigma-Aldrich | M1404 | |
| Chemical compound, drug | CK666 | Sigma-Aldrich | SML0006-5MG | |
| Chemical compound, drug | SMIFH2 | Wako | 4401/10 | |
| Chemical compound, drug | Blebbistatin | Sigma | B0560-1MG | |
| Chemical compound, drug | Rho Activator II | Cytoskeleton, Inc | Cat. #CN03 | |
| Chemical compound, drug | Alexa Fluor 568 phalloidin | Invitrogen | A12380 | 1:400 |
| Chemical compound, drug | Alexa Fluor 488 phalloidin | Invitrogen | A12379 | 1:500 |
| Software, algorithm | ImageJ Fiji | *Schindelin et al., 2012* | Version 2.14.0 | |
| Software, algorithm | Trim Galore | *Krueger et al., 2023* | Version 0.6.10 | |
| Software, algorithm | HISAT2 | *Kim et al., 2019* | Version 2.2.1 | |
| Software, algorithm | Samtools | *Danecek et al., 2021* | Version 1.9 | |
| Software, algorithm | Stringtie | *Kovaka et al., 2019* | Version 2.2.1 | |
| Software, algorithm | PIVlab | *Thielicke and Sonntag, 2021* | Version 2.56 | |
| Software, algorithm | MATLAB | Mathworks | R2024a | |
| Software, algorithm | FreeFEM++ | *Hecht, 2012* | Version 4.15 | |

## Cell cultures and transfection

Caco-2 (ATCC) cells were cultured in DMEM/Ham's F-12 (FUJIFILM Wako Pure Chemical Corporation, 048-29785) supplemented with 10% fetal bovine serum (Sigma, F7524, Lot. BCBV 4600) and 1%

penicillin/streptomycin (nacalai, 26253-84) at 37°C, 5% $CO_2$ on collagen type I coated Dish (60 mm, IWAKI, 4010-010). For live imaging of actin, we used Lifeact-RFP-transfected Caco-2 cells which were established from Caco-2 (ATCC) in *Ozawa et al., 2020*. To live-image the microtubule dynamics, we transiently transfected Caco-2 cells with EMTB-3XGFP using Lipofectamine LTX Reagent with PLUS Reagent (Invitrogen, 15338100), according to the manufacturer's protocol. EMTB-3XGFP was a gift from William Bement (*Miller and Bement, 2009*) (Addgene plasmid # 26741; http://n2t.net/addgene: 26741; RRID:Addgene_26741) For Lifeact-mEmerald (pLVSIN-EF1a-Lifeact-mEmerald-IRES-pur), mEmerald-Lifeact-7 was inserted into pLVSIN-EF1a-IRES-pur at the BamHI and NotI sites by In-Fusion. The construction of pLVSIN-EF1a-IRES-pur has been described previously (*Nakamura et al., 2012*). mEmerald-Lifeact-7 was a gift from Michael Davidson (Addgene plasmid # 54148; http://n2t.net/addgene:54148; RRID:Addgene_54148).

For protein depletion, cells were transfected with Stealth siRNA or Silencer Select siRNA using Lipofectamine RNAi MAX (Invitrogen). The following Stealth siRNAs were used: MYH9HSS106871 for Myosin IIA; MYH10HSS106875 for Myosin IIB; DAAM1HSS177085 for DAAM1; DIAPH2HSS102773 for DIAPH2; Negative Control Med GC Duplex (#10002823) for negative control. The following Silencer Select siRNA was used: VCL s14764 for vinculin. We examined the effects of RNA interference at 5 days.

## Immunoblotting

Cells were collected in RIPA buffer containing 1× cOmplete EDTA-free protease inhibitor (Roche, 05056489001) and were then mechanically lysed by passage through a 27 G needle on ice. Samples were boiled with 10% 2-mercaptoethanol for 3 min and separated by SuperSep 7.5% SDS–polyacrylamide gel electrophoresis (Fujifilm, 198–14941) at 250 V/30 mA for 70 min, and subsequently transferred to a 0.45-µm pore size PVDF membrane (Cytiva, 1080682) at 250 V/300 mA for 90 min in an ice bath. The PVDF membrane was blocked with Blocking One (Nacalai Tesque, 03953-95) at 4°C overnight and incubated with appropriate primary antibodies for 60 min at room temperature (RT). After washing the membrane with Tris-buffered saline with 0.1% Tween 20 (TBST), it was incubated with appropriate HRP-conjugated secondary antibodies for 60 min at RT. The membrane was then washed with TBST, followed by enhanced chemiluminescence detection (Bio-Rad, 1705060) using LAS-3000 mini (Fujifilm) for image acquisition. Signal intensity was analyzed using the Gel Analysis tool in ImageJ Fiji (*Schindelin et al., 2012*). The following antibodies were used: rabbit anti-Myosin IIA (Sigma-Aldrich, M8064, dilution 1:1000); rabbit anti-Myosin IIB (Cell Signaling Technology, 8824, dilution 1:1000); mouse anti-GAPDH (Santa Cruz Biotechnology, sc-166574, dilution 1:1000); HRP-conjugated goat anti-rabbit IgG (Invitrogen, T20926, dilution 1:1000); HRP-conjugated goat anti-mouse IgG (Invitrogen, T20912, dilution 1:1000). As a protein ladder marker, Precision Plus Protein Dual Color Standard (Bio-Rad, 1610374) was used.

## Live cell imaging

For live-image data in *Figures 1 and 2*, *Figure 2—figure supplement 1*, we used an inverted fluorescence microscope (Olympus, IX-81) equipped with a spinning disk confocal imaging unit (Yokogawa, CSU-X1), a 60×/1.35 oil immersion objective (Olympus, UPLSAPO60XO), and a 561-nm laser (Coherent, Sapphire LP) for RFP excitation or a 488-nm laser (Coherent, Sapphire LP) for GFP excitation. We used a 40×/1.35 oil-immersion objective (Olympus, UApo/340) for the microtubule snapshots in *Figure 2—figure supplement 1*. The cells were seeded sparsely on a collagen type I coated glass-based dish (IWAKI, 4970-011), and incubated in a stage-top incubator (Tokai Hit) at 37°C with 5% $CO_2$ during live imaging.

We took fluorescence images with multiple z-stacks (number of slices: 7 and $\Delta z$: 0.5 µm) by EMCCD (Andor Technology, iXon+) every 15 min, and then made maximum intensity Z projections. For the microtubule snapshots in *Figure 2—figure supplement 1*, we applied a different condition (number of slices: 5 and $\Delta z$: 1 µm).

For the inhibitor experiments, the following inhibitors were used: latrunculin A (Sigma, L5163-100UG); Jasplakinolide (Toronto Research Chemicals Inc, J210700); Nocodazole (Sigma-Aldrich, M1404); CK666 (Sigma-Aldrich, SML0006-5MG); SMIFH2 (Wako, 4401/10); Blebbistatin (Sigma, B0560-1MG); Rho Activator II (Cytoskeleton, Inc, Cat. #CN03). We started live imaging about 2–3 hr after seeding cells and added the inhibitors about 40 min before the live imaging.

When we observed the dynamics of the beads attached to the dorsal membrane of the cells, we used 2 µm carboxylate-modified beads (Invitrogen, F8887), and live-imaged the dynamics using the DIC channel of the microscope (Olympus, IX-81).

## Analysis of rotation of nucleus

We quantified the rotational behaviors of cells by manually tracking the dynamics of two nucleoli of each cell on DIC images (Segmentation Editor, Fiji). We defined the rotational angle of the cell nucleus by that of the line connecting the two nucleoli and analyzed the rotational dynamics using Python. In *Figure 1A, B*, we defined the initial time point of the measurement of the nuclear rotation by the time when we started the live imaging. In the inhibitor experiments in *Figure 2*, we determined the initial time point of the measurement as the time 5 hr after the initiation of live imaging, accounting for the time lag needed for the inhibitors to exert their effects. For each experimental condition, single-cell data were collected from multiple cells in a single culture dish; the number of cells analyzed is given in each figure legend.

## Immunofluorescence antibody staining and microscopy

Cells were seeded on collagen type I coated cover slips (Neuvitro Corporation, NEU-H-12-COLLAGEN-45) and treated with the inhibitors. After 8 hr, the cells were fixed with 2% PFA in PBS(−) for 10 min, permeabilized with 0.25% Triton X-100 in PBS(−) for 10 min, blocked with 3% BSA in PBS(−) for 30 min. Then, we incubated cells with primary antibodies (2 hr), secondary antibodies (1 hr), and phalloidin (30 min) in a blocking buffer (3% BSA in PBS(−)). After washing with PBS(−) three times, the samples were mounted with a mounting medium with DAPI (Vector Laboratories, VECTASHIELD, H-1200). All the processes were performed at RT.

We used rabbit anti-Myosin IIA (Sigma-Aldrich, M8064, 1:1000 for IF), mouse anti-Vinculin (Sigma, V9131, 1:200 for IF), and mouse anti-Ezrin (Abcam, ab4069, 1:1000 for IF) as the primary antibodies and Alexa Fluor 488 goat anti-rabbit IgG (Sigma-Aldrich, 11034, 1:1000 for IF), Alexa Fluor 488 goat anti-mouse IgG (Invitrogen, A11029, 1:1000 for IF), and Alexa Fluor 647 donkey anti-mouse IgG (Sigma-Aldrich, AP192SA6, 1:1000 for IF) as the secondary antibodies, respectively. For actin staining, we used Alexa Fluor 568 phalloidin (Invitrogen, A12380, 1:400).

To analyze the sample, we took fluorescence images with multiple z-stacks ($\Delta z$: 0.32 µm) using a laser scanning confocal microscope (Zeiss, LSM880) equipped with Plan-Apochromat 63×/1.4 Oil DIC M27. Images were processed with Fiji.

## RNA-seq analysis

RNA was extracted from three independent samples of Caco-2 cells using the RNeasy Kit (QIAGEN). Libraries were sequenced on an Illumina NextSeq 2000 platform with 100 bp single-end reads, supported by the RIKEN BDR DNA Analysis Facility at the Laboratory for Developmental Genome System. Adapter sequences were removed using Trim Galore. Reads were mapped to the reference genome (grch38) using HISAT2, sorted using samtools, and transcript abundances were quantified as transcripts per million using StringTie.

## Expansion microscopy

Protein-retention ExM was carried out as described previously (*Zhang et al., 2020*). Cells were cultured on collagen type I (Sigma, C8919-20ML) coated cover slips and treated with the inhibitors. Fixation, permeabilization, and blocking were performed as described above. To visualize F-actin, the cells were stained with Alexa Fluor 488 phalloidin (Invitrogen, A12379, 1:400) and next stained for 60 min with a rabbit anti-Alexa Fluor 488 antibody (abcam, ab150077, 1:500) as the primary antibody and Alexa Fluor 488-conjugated goat anti-rabbit IgG (1:1000) as the secondary antibody (*Park et al., 2020*). After washing, cells were incubated with 100 µg/mL of 6-((Acryloyl)amino)hexanoic acid, succinimidyl ester overnight at RT in the dark. Samples were incubated in gelation solution (8.6% (wt/wt) sodium acrylate, 2.5% (wt/wt) acrylamide, 0.15% (wt/wt) *N,N'*-methylenebisacrylamide, 2 M NaCl, 1× PBS, 0.1% TEMED, 0.1% ammonium persulfate) for 5 min on ice. Gelation was allowed to proceed at RT for 1 hr. The gel and a cover slip were removed with tweezers and incubated with digestion buffer (0.5% Triton X-100, 1× TE buffer, 1 M NaCl, 8 unit/ml Proteinase K) overnight at RT in the dark. The gels were removed from the digestion buffer and placed in 50 ml of Milli-Q water. Water was

exchanged three times every 30 min. Most of the gels expanded to about 4.5 times their original size. Gels were placed on a poly-L-lysine (Sigma-Aldrich, P4707) coated glass bottom dish, and fluorescence images were taken by a laser scanning confocal microscope (Zeiss, LSM880).

## LLSM and image processing

The LLSM was home-built in the Kiyosue laboratory at RIKEN Center for Biosystems Dynamics Research following the design of the Betzig laboratory (*Chen et al., 2014*) under a research license agreement from Howard Hughes Medical Institute. Electric wiring was performed at RIKEN Advanced Manufacturing Support Team. Metal parts were processed by Maeda Precision Manufacturing Ltd and Zera Development Co. To create a lattice light sheet, a dithered square lattice was used through a spatial light modulator (Fourth Dimension Displays) in combination with an annular mask with 0.55 out and 0.44 inner numerical apertures (Photo-Sciences) and a custom NA 0.65 excitation objective (Special Optics). Images were acquired through a CFI Apo LWD 25XW 1.1-NA detection objective (Nikon) and a scientific sCMOS camera, Orca Flash 4.0 v3 (Hamamatsu Photonics). Caco-2 cells expressing Lifeact-mEmerald were seeded on a collagen-coated coverslip 3 hr before imaging. During imaging, cells were maintained in DMEM/Ham's F-12 (FUJIFILM Wako Pure Chemical Corporation, 048-29785) supplemented with 10% fetal bovine serum (Sigma, F7524, Lot. BCBV 4600) at 37°C, 5% $CO_2$. For live imaging of Lifeact-mEmerald, a 488-nm laser (MPB Communications) and a long-pass emission filter BLP01-488R-25 (Semrock) were used. Image stacks were collected with a 200-nm step size between planes with 10 ms per plane exposure time and 14.8-s time interval. After the acquisition, images were deskewed and deconvolved using LLSpy. After deskew processing, the voxel pitch was 0.104 × 0.104 × 0.103 μm.

## PIV analysis

PIV analysis was performed using PIVlab (*Thielicke and Sonntag, 2021*) for the time-lapse images obtained by LLSM. The velocity vector fields were calculated using a multi-grid interrogation (64 × 64, 32 × 32, and 16 × 16 pixel sizes of interrogation windows with 50% overlap each). PIV was performed in the masked area, and the masked area was determined by thresholding the time-integrated image. Using the velocity vector field $(v_x, v_y)$, we first calculated azimuthal $(v_\varphi)$ and radial $(v_\rho)$ velocities as $v_\varphi = \hat{x} v_y - \hat{y} v_x$ and $v_r = \hat{x} v_x + \hat{y} v_y$, where $\hat{x} = x/r$ and $\hat{y} = y/r$ with $(x, y)$ being the position from the center of cell and $r = \sqrt{x^2 + y^2}$. Here, the cell center was taken as the $xy$-coordinate of the highest position of the unimodal-shaped cell. From the DIC image, the nucleus always rotates around the center of the nucleus, with the highest position of the cell just above the nucleus center. Then, the angular velocity $\omega$ was obtained as $\omega = v_\varphi/r$. The temporal averages of $v_\varphi$ and $\omega$ are shown in *Figure 6— figure supplement 1* for all samples analyzed control cells (DMSO) (*Figure 6—figure supplement 1A–D*) and cells treated with SMIFH2 (*Figure 6—figure supplement 1E–H*). The angular averages of $v_\varphi$ (*Figure 6—figure supplement 1I*), $v_r$ (*Figure 6—figure supplement 1J*), and $\omega$ (*Figure 6C*) were obtained from their temporal averages at each spatial point. These averages were plotted against the radius scaled by the inner radius of the actomyosin ring in *Figure 6D*. We here manually identified the inner edge of the actomyosin ring and calculated the inner radius for each cell.

## Theoretical model

We here describe a derivation of our 3D model (*Fürthauer et al., 2012*). We assume a low Reynolds number limit, a steady state, and incompressibility. In the theory of active chiral fluid, the momentum conservation is represented as:

$$0 = \partial_\beta(\sigma^s_{\alpha\beta} + \sigma^e_{\alpha\beta} + \sigma^a_{\alpha\beta}), \tag{2}$$

where $\sigma^s_{\alpha\beta}$ and $\sigma^a_{\alpha\beta}$ are the symmetric and asymmetric parts of the deviatoric stress, respectively. $\sigma^e_{\alpha\beta}$ is Ericksen stress (hydrostatic stress). The indices α, β, and γ denote the three Cartesian coordinates $x, y$, and $z$. The constitutive equations of the deviatoric stress are given:

$$\sigma^s_{\alpha\beta} = 2\eta u_{\alpha\beta} + \zeta^a p_\alpha p_\beta, \tag{3}$$

$$\sigma^a_{\alpha\beta} = 2\eta'(\Omega_{\alpha\beta} - \omega_{\alpha\beta}), \tag{4}$$

where $u_{\alpha\beta} = (\partial_\alpha v_\beta + \partial_\beta v_\alpha)/2$ and $\omega_{\alpha\beta} = (\partial_\alpha v_\beta - \partial_\beta v_\alpha)/2$ is the strain rate and the vorticity. $\Omega_{\alpha\beta}$ is the spin rotation rate describing the intrinsic rotation rate of local volume elements. $\eta$ and $\eta'$ are viscosity coefficients, and $\zeta^a$ is a coefficient of the achiral active stress. We here only consider anisotropic contributions of active terms allowed in a chiral nematic active fluid for simplicity. Also, in this study, since we assume that the orientational field $\mathbf{p}$ is fixed to be a concentric pattern, we omit the terms that derive from the molecular field. We thus define $\sigma^e_{\alpha\beta} = -P\delta_{\alpha\beta}$, where $P$ is the pressure serving as a Lagrange multiplier to satisfy the incompressibility.

The angular momentum conservation is given by the following equation:

$$\partial_\gamma M_{\alpha\beta\gamma} = 2\sigma^a_{\alpha\beta}, \tag{5}$$

where $M_{\alpha\beta\gamma}$ is the angular momentum flux. The constitutive equation is written as:

$$M_{\alpha\beta\gamma} = \kappa \partial_\gamma \Omega_{\alpha\beta} + \zeta^c \epsilon_{\alpha\beta\delta} p_\delta p_\gamma, \tag{6}$$

where $\kappa$ is a dissipative coefficient and $\zeta^c$ is a coefficient of the active chiral stress which reflects the symmetry of the torque dipole represented in *Figure 7*, which is called nematic chiral rod motor (*Fürthauer et al., 2012*). $\epsilon_{\alpha\beta\gamma}$ is the Levi-Civita symbol.

We derive the following equation of motion from *Equations 2–6*,

$$\begin{aligned}
0 = & -\partial_\alpha P + 2\eta\partial_\beta u_{\alpha\beta} + \partial_\beta \zeta^a p_\alpha p_\beta + \frac{1}{2}\partial_\beta\partial_\gamma \zeta^c \epsilon_{\alpha\beta\delta} p_\delta p_\gamma \\
& + \frac{\kappa}{4\eta'}\partial_\gamma^2 \left(\partial_\beta P\delta_{\alpha\beta} - 2\eta\partial_\beta u_{\alpha\beta} - 2\partial_\beta \zeta^a p_\alpha p_\beta + 2\eta'\partial_\beta \omega_{\alpha\beta}\right).
\end{aligned} \tag{7}$$

In the final term of *Equation 7*, the length scale $\ell = \sqrt{\kappa/\eta'}$ is a characteristic molecular scale. Since we consider the hydrodynamics at the cell scale, we take the limit of $\ell \to 0$ and omit the final term. Finally, applying the incompressibility condition $\partial_\gamma v_\gamma = 0$, we obtain the final form:

$$0 = -\partial_\alpha P + \eta\partial_\gamma^2 v_\alpha + \partial_\beta \zeta^a p_\alpha p_\beta + \frac{1}{2}\partial_\beta\partial_\gamma \zeta^c \epsilon_{\alpha\beta\delta} p_\delta p_\gamma, \tag{8}$$

which is equivalent to *Equation 1*. By introducing non-dimensional velocity, position, and pressure as $\tilde{\mathbf{v}} = \frac{\eta}{\zeta^a R_0}\mathbf{v}$, $(\tilde{x}, \tilde{y}, \tilde{z}) = (x/R_0, y/R_0, z/R_0)$ and $\tilde{P} = P/\zeta^a$, where $R_0$ is the cell radius, the non-dimensional form of *Equation 1* is given by

$$0 = -\tilde{\nabla}\tilde{P} + \tilde{\nabla}^2\tilde{\mathbf{v}} + \tilde{\nabla}\cdot\mathbf{pp} + \frac{1}{2}\zeta\tilde{\nabla}\times\tilde{\nabla}\cdot\mathbf{pp} \tag{9}$$

with the non-dimensional parameter

$$\zeta = \frac{\zeta^c}{\zeta^a R_0}, \tag{10}$$

which essentially describes the chiral activity relative to the non-chiral active contribution. Hereafter, we omit tilde from the non-dimensional velocity, position, and pressure.

In the numerical simulations, for simplicity, we suppose that the cell is axisymmetric as shown in *Figure 7*, *Figure 7—figure supplement 1*. Based on the experimental observations, we consider that the actomyosin bundles align along the circumferential direction: the concentric pattern of the actomyosin ring. We here represent the orientational order $\mathbf{p}$ of the actomyosin in the cylindrical coordinate $(\rho, \varphi, z)$. Since $\mathbf{p}$ is aligned in the circumferential direction, $\mathbf{p}(\rho, z)$ is given in the cylindrical coordinate by

$$\mathbf{p}(\rho, z) = S(\rho, z)\left(0, 1, 0\right)^t, \tag{11}$$

where $S(\rho, z)$ is the effective strength of the orientation of the actomyosin and takes a finite value in the domain where the actomyosin ring is present. Since we did not see any specific orientational order in the direction of the cell height, at least at our imaging resolution, we considered the orientation of the actomyosin bundle to be parallel to the substrate, and the $z$ component of $\mathbf{p}(\rho, z)$ is zero. In the

cylindrical coordinate, the equation of motion *Equation 1* for the fluid velocity $\mathbf{v} = (v_\rho, v_\varphi, v_z)^t$ and the pressure $P$ read

$$\partial_\rho^2 v_\varphi + \partial_z^2 v_\varphi + \frac{1}{\rho}\partial_\rho v_\varphi - \frac{1}{\rho^2}v_\varphi = \frac{\zeta}{\rho}S\partial_z S, \tag{12}$$

$$\partial_\rho^2 v_\rho + \partial_z^2 v_\rho + \frac{1}{\rho}\partial_\rho v_\rho - \frac{1}{\rho^2}v_\rho = \partial_\rho P + \frac{S^2}{\rho}, \tag{13}$$

$$\partial_\rho^2 v_z + \partial_z^2 v_z + \frac{1}{\rho}\partial_\rho v_z = \partial_z P, \tag{14}$$

$$\frac{v_\rho}{\rho} + \partial_\rho v_\rho + \partial_z v_z = 0. \tag{15}$$

We numerically solve the equations with the following boundary shape, as shown in *Figure 7—figure supplement 1*. The dorsal boundary of the cell is well described by the following equations:

$$\begin{cases} z & = Z_0 - r_1 + \sqrt{r_1^2 - \rho^2} \quad (0 \leq \rho \leq \rho_1) \\ z & = Z_2 - \sqrt{r_2^2 - (\rho - \rho_2)^2} \quad (\rho_1 \leq \rho \leq \rho_3) \\ z & = -(\tan\alpha)(\rho - R_0) \quad (\rho_3 \leq \rho \leq R_0) \end{cases} \tag{16}$$

with

$$\begin{cases} \rho_1 & = \dfrac{r_1\rho_2}{r_1 + r_2} \\ \rho_3 & = \rho_2 - r_2\sin\alpha \\ r_2 & = (\sin\alpha)(\rho_2 - R_0) + (\cos\alpha)Z_2 \\ Z_2 & = -\dfrac{r_1 - Z_0 - (\cos\alpha)(r_1 + (\rho_2 - R_0)\sin\alpha) + \sqrt{(-r_1 + (r_1 - Z_0)\cos\alpha + R_0\sin\alpha)(-r_1 + (r_1 - Z_0)\mathrm{c}}}{\sin^2\alpha} \end{cases}$$

(17)

where $Z_0$ is the cell height, $R_0$ is the cell radius, $r_1$, $\rho_2$, and $\alpha$ are the parameters that determine the cell shape. We determined the values of those parameters from the experimental data as $Z_0 = 8.2\ \mu m$, $r_1 = 15.0\ \mu m$, $\rho_2 = 22.0\ \mu m$, $R_0 = 35.0\ \mu m$, and $\alpha = 7.0 \times 2\pi/360.0$. The ventral boundary is specified by $z = 0$.

Since the actomyosin ring is located along the dorsal surface, we practically consider that as shown in *Figure 7B* $S(\rho, z)$ is given by

$$S = \left(\frac{1}{2}\tanh\left(\lambda_1\left((\sin\beta)\rho + (\cos\beta)z - (\sin\beta)R_0 + \xi\right)\right) + \frac{1}{2}\right)\left(\frac{1}{2}\tanh\left(-\lambda_2\left((\cos\beta)\rho - (\sin\beta)z - \xi_3\right)\right) + \frac{1}{2}\right) \tag{18}$$

in the range $(\cos\beta)\rho - (\sin\beta)z - \frac{\rho_3'}{\cos\beta} + (\sin\beta)(\tan\beta)R_0 \leq 0$, and

$$S = \frac{1}{2}\tanh\left(\lambda\left(r_2' + \xi - \sqrt{(\rho - \rho_2)^2 + (z - Z_2')^2}\right)\right) + \frac{1}{2} \tag{19}$$

in the range $(\cos\beta)\rho - (\sin\beta)z - \frac{\rho_3'}{\cos\beta} + (\sin\beta)(\tan\beta)R_0 > 0$. Here, $\rho_3'$, $r_2'$ and $Z_2'$ are given by

$$\begin{cases} \rho_3' & = \rho_2 - r_2'\sin\beta \\ r_2' & = (\sin\beta)(\rho_2 - R_0) + (\cos\beta)Z_2' \\ Z_2' & = -\dfrac{r_1 - Z_0' - (\cos\beta)(r_1 + (\rho_2 - R_0)\sin\beta) + \sqrt{(-r_1 + (r_1 - Z_0')\cos\beta + R_0\sin\beta)(-r_1 + (r_1 - Z_0')\mathrm{c}}}{\sin^2\beta} \end{cases}$$

(20)

where $\lambda_1$, $\lambda_2$, $\beta$, $\xi$, $\xi_3$, and $Z_0'$ are the parameters that determine the distribution of activity inside the cell. We used the following parameter values, $\lambda_1 = 150.0/R_0 \, \mu\mathrm{m}^{-1}$ , $\lambda_2 = 20.0/R_0$ , $\beta = 1.7 \times \alpha$, $\xi = 0.06 \times R_0 \, \mu\mathrm{m}$, $\xi_3 = 0.7 \times R_0 \, \mu\mathrm{m/s}$ and $Z_0' = 1.5 \times Z_0$.

We numerically solved the equations of motion by assuming the no-slip boundary condition for the ventral boundary, the free slip boundary condition for the dorsal surfaces (*Kashiwabara et al., 2016*), and the vanishing flow velocity for $v_\rho$ and $v_\varphi$ and the continuity for $v_z$ at the cell center $\rho = 0$. We do not include any organelles such as a nucleus in the model for simplicity. The equations were solved numerically with a finite element method using software FreeFEM++ (*Hecht, 2012*).

We determined the parameter values to reproduce the experimental observation as follows. As shown in *Figure 7—figure supplement 2A–H*, the peak values of the radial and azimuthal velocities are comparable. To reproduce this property, we set the parameter value of $\zeta$ to be $\zeta = 0.004$. The simulation also reproduced the shift in the peak positions between two velocities. The peak values of the velocities in the simulation were about $2 \times 10^{-4}$ (unitless), while those in the experiments were about $5 \times 10^{-3} \sim 10^{-2} \, \mu\mathrm{m/s}$, giving the velocity scale to be $\frac{\zeta^a R_0}{\eta} = 50 \, \mu\mathrm{m/s}$ and the time scale to be $\frac{\eta}{\zeta^a} = 0.7$ s. With this time scale, the peak value of the angular velocity was about 80 degree/hr as shown in *Figure 7H*, consistent with the experimental observation shown in *Figure 6C, D*.

## Quantification of dorsal and ventral actomyosin

Fluorescent signals of anti-Myosin IIA and phalloidin were obtained by LSM880 (Zeiss) with Airyscan and processed by ImageJ Fiji. The obtained images were resliced and ten *x–z* slices containing the cell center were processed with mean intensity projection. The dorsal and ventral surfaces were manually traced with 10-pixel width, and the average signal intensities in the traced regions were quantified. The cell edge regions of overlapping dorsal and ventral traces were annotated as 'peripheral region' and excluded from the quantification.

## Acknowledgements

This work was supported by Kakenhi grant 19K16096 (TY) 23K14186 (TI), 22H05170, 23H02455 (TS), JST ACT-X Grant JPMJAX2423 (TI), and JST CREST Grant JPMJCR1852 (TS) and the core funding at RIKEN Center for Biosystems Dynamics Research. TY and TI were supported by Grant-in-Aid for JSPS Fellows 18J01239 (TY) and 22KJ3145 (TI). We thank H Saito, T Kato, and H Hamada for the advice and support during experiments, M Hayakawa, G Ogita, B Bhattacherjee, R Nishizawa, K Kawaguchi, K Adachi, Y Fukai, R Cerbus, T Kato, S Fürthauer, YH Tee, and AD Bershadsky for fruitful discussions, W Kimura for sharing reagents, and the Laboratory for Developmental Genome System for support with the RNA-seq analysis.

## Additional information

### Funding

| Funder | Grant reference number | Author |
| --- | --- | --- |
| Japan Society for the Promotion of Science | 19K16096 | Takaki Yamamoto |
| Japan Society for the Promotion of Science | 23K14186 | Tomoki Ishibashi |
| Japan Society for the Promotion of Science | 22H05170 | Tatsuo Shibata |
| Japan Society for the Promotion of Science | 23K27148 | Tatsuo Shibata |
| Core Research for Evolutional Science and Technology | 10.52926/JPMJCR1852 | Tatsuo Shibata |

| Funder | Grant reference number | Author |
| --- | --- | --- |
| RIKEN Center for Biosystems Dynamics Research | | Tatsuo Shibata |
| Japan Society for the Promotion of Science | 18J01239 | Takaki Yamamoto |
| Japan Society for the Promotion of Science | 22KJ3145 | Tomoki Ishibashi |
| Japan Science and Technology Agency | JPMJAX2423 | Tomoki Ishibashi |

The funders had no role in study design, data collection, and interpretation, or the decision to submit the work for publication.

## Author contributions

Takaki Yamamoto, Tomoki Ishibashi, Conceptualization, Formal analysis, Funding acquisition, Investigation, Methodology, Writing – original draft, Writing – review and editing; Yuko Mimori-Kiyosue, Mitsusuke Tarama, Investigation, Writing – review and editing; Sylvain Hiver, Naoko Tokushige, Investigation; Masatoshi Takeichi, Conceptualization, Investigation, Writing – review and editing; Tatsuo Shibata, Conceptualization, Formal analysis, Supervision, Funding acquisition, Investigation, Methodology, Writing – original draft, Writing – review and editing

## Author ORCIDs

Takaki Yamamoto (ID) https://orcid.org/0000-0002-4321-5269
Tomoki Ishibashi (ID) https://orcid.org/0000-0001-6652-9343
Mitsusuke Tarama (ID) https://orcid.org/0000-0002-2708-1774
Masatoshi Takeichi (ID) https://orcid.org/0000-0002-9931-3378
Tatsuo Shibata (ID) https://orcid.org/0000-0002-9294-9998

## Decision letter and Author response

Decision letter https://doi.org/10.7554/eLife.102296.sa1
Author response https://doi.org/10.7554/eLife.102296.sa2

## Additional files

### Supplementary files

MDAR checklist

### Data availability

Source data for all graphs and original code for the numerical integration of the model have been deposited and are publicly available at https://doi.org/10.5281/zenodo.8254364.

The following dataset was generated:

| Author(s) | Year | Dataset title | Dataset URL | Database and Identifier |
| --- | --- | --- | --- | --- |
| Yamamoto T, Ishibashi T, Hiver S, Tokushige N, Tarama M, Takeichi M, Shibata T | 2025 | Epithelial cell chirality emerges through the dynamic concentric pattern of actomyosin cytoskeleton | https://doi.org/10.5281/zenodo.8254364 | Zenodo, 10.5281/zenodo.8254364 |

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
