## [Editor Report]

Although the actomyosin cytoskeleton has been shown to play an important role, the principles by which molecular chirality leads to the chirality of cells, tissues, and organs remain largely unexplored. This important study reveals that the concentric actomyosin network at the apical side of Caco-2 cells, rather than the ventral chiral stress fibers, drives the rotational movement of the nucleus and cytoplasmic flow in the same direction. The convincing data are supported by a theoretical model based on the theory of active fluids, which explains how unidirectional rotation at the cellular scale can arise from the chirality of actomyosin filaments at the molecular scale, even in the absence of chiral orientational order at the cellular scale.

---

## [Decision Letter]

[Editors' note: this paper was reviewed by Review Commons.]

---

## [Author Response]

We are grateful to all the reviewers who took their time to evaluate this work. We have carefully revised the manuscript according to the comments. In the following, we give point-by-point answers to the issues raised.

Reviewer #1 Major points1. It is important to provide information on whether the chiral motion of the nucleus is sensitive to the cell's physical environment. For example, does the motion still occur when the cells are plated on substratum with different physicochemical properties, or when the cells form a monolayer?

We appreciate the reviewer’s comment and have elucidated further the conditions under which the chiral rotational motion occurs. In the previous manuscript, cells were plated on collagen-coated glass substrates. According to the reviewer’s suggestion, we tested other substrates, such as the glass surfaces coated with fibronectin and poly L-lysine. When cells were cultured on fibronectin-coated glass, they exhibited a similar clockwise nuclear rotation as observed on collagen-coated substrates, but the rotation speed was slightly reduced (Figure 1—figure supplement 1). On poly-L-lysine-coated or uncoated glass, the speed was further diminished, and some cells hardly rotated at all (Figure 1—figure supplement 1). Notably, while different coatings influence the speed of nuclear rotation, they do not alter its direction. We included this result in the revised manuscript as below:

“The rotational speed of the nucleus depended on the type of coating applied to the glass substrate, although the direction of rotation remained unaffected. When cells were cultured on fibronectin-coated glass, they exhibited the same clockwise nuclear rotation as observed on collagen-coated substrates, albeit at a slightly reduced speed (Figure 1—figure Supplement 1). On poly-L-lysine-coated or uncoated glass, the rotation speed further decreased, with some cells exhibiting little rotation (Figure 1— figure Supplement 1). Notably, while different coatings influenced the speed of nuclear rotation, they did not alter its direction.”

Concerning whether chiral motion still occurs in monolayers, we have already found that nuclear rotation does not happen in the confluent cultures of Caco-2 cells. Since this is an important point, we will further study what cellular conditions affect nuclear motion and publish the results in future papers rather than mentioning this observation in the current manuscript.

2. The characterization of the concentric actomyosin rings requires further clarification. Firstly, the organization of the actomyosin cytoskeleton along the z-axis is not clearly demonstrated in the figures. For example, in the images presented in Figure 3, it is unclear which structures are localized to the ventral cell cortex and which ones to the dorsal cortex. Including a montage demonstrating the actin and myosin organization at different z-planes would be helpful. Secondly, the terms 'thick' and 'thin' actin filaments are not clearly defined. Finally, the authors should test whether the concentric actin rings are under tension (e.g., by laser ablation). This experiment will help demonstrate that the myosin and actin located in the concentric rings form functional contractile units, which is important because the colocalization of thin actin and myosin filaments could not be directly confirmed using expansion microscopy.

We appreciate the important comment on the different organization of the actomyosin cytoskeleton shown in Figure 4 in the revised manuscript (which was Figure 3 in the original manuscript). In revision, we have made composite images in which the dorsal and ventral cortex are shown in different colors to distinguish between them (Figure 4A’ and C’ in the revised manuscript).

With respect to the “thick” and “thin” actin filaments, we have decided not to use these terms. Instead, we used “stress fiber” and “dorsal actin fibers” to distinguish between these two populations of actin. The stress fibers are located at the ventral side, anchored to focal adhesions at their termini, showing a dextral swirling pattern (red color in Figure 4A’). In contrast, the dorsal actin fibers are located along the dorsal membrane, and their orientations tend to be parallel to the tangential direction of the cell edge, forming a concentric pattern (green in Figures 4A’ and C’).

With respect to the colocalization of actin and myosin along the dorsal membrane, the images taken by confocal microscopy in Figure 3 in the revised manuscript show a good colocalization between the two components. In particular, both actin and myosin form concentric patterns in the cells treated with SMIFH2, which are well colocalized (Figure 3B). Thus, we consider that the dorsal actomyosin fibers are functional contractile units.

3. The authors presented several pieces of evidence supporting the role of concentric actin rings in driving the motion of the cytoplasm and nucleus. Firstly, the concentric actin rings exhibit chiral motion in the same direction as the nucleus, with the velocity of this motion peaking in the region of the rings. Secondly, treating cells with SMIFH2 increased the actomyosin ring signal, which correlated with an elevated speed of nuclear rotational motion. Finally, expressing a Rho activator, which disrupts the formation of concentric actomyosin rings, prevents nuclear rotation. However, the increase in rotational velocity in SMIFH2-treated cells was rather mild, and the expression of the Rho activator caused drastic changes in the overall organization of the actin cytoskeleton, raising questions about the specificity of this treatment. These concerns can be addressed if the authors further test the role of the concentric actin rings in driving chiral movement by directly perturbing the dorsal actin rings, such as through laser microdissection.

We appreciate the insightful comment on the role of the concentric actin rings at the dorsal side. We agree with the reviewer that to obtain direct evidence that these rings drive the cytoplasmic motion and nucleus rotation, it would be beneficial to observe whether these rotations cease upon removal of the dorsal actin fibers. Laser ablation would be one possibility to directly perturb them. However, we concluded that such an experiment is not feasible for two reasons. First, the z distance between the dorsal actin rings and the ventral stress fibers is less than 0.5 µm, according to the LLSM observations (Figure 5). Considering the resolution of the microscope in the z-direction, it could be difficult to destroy only the dorsal actin rings selectively. Second, after we could disrupt a small area of the dorsal actin fibers, no matter how slow, it will recover in about 10 minutes. The rotational motion of the cytoplasm and nucleus is slow. The angular speed of the nucleus is about 50 degrees/hour on average. In 10 minutes, the nucleus would rotate in less than 10 degrees if the dorsal fibers had not been destroyed. It would be difficult to determine whether such a small rotational motion stopped or continued upon the laser ablation. Conducting experiments to perturb only the dorsal actin rings is a future challenge.

4. The mathematical model predicts that the cytoplasm near the dorsal concentric actomyosin rings undergoes inward sinistral rotation, while the ventral cytoplasm undergoes outward dextral rotation. The ventral outward dextral flow is inferred from the orientation of the ventral stress fibers; however, the flow has not been directly demonstrated. Additionally, the orientation of the stress fibers in SMIFH2-treated cells seems to indicate sinistral (clockwise) rotation, rather than dextral (counter-clockwise) rotation. Is it possible to directly monitor the motion of the ventral cytoplasm, for example, by tracking the movement of ventrally localized organelles or particles, such as exogenous GEMs?

We acknowledge the reviewer’s important comment on the cytoplasmic flow profile. Every radial fiber in SMIFH2-treated cells associates with a focal adhesion at one end (Figure 1—figure supplement 2). According to Figure 4C, its free end internally extends in a clockwise direction along the dorsal membrane (see the change in the color along the radial fibers in Figure 4C), where the inward flow is present. Therefore, the sinistral pattern of the radial fiber in SMIFH2-treated cells is consistent with the theoretical prediction.

The outward flow at the ventral side predicted by the theoretical model is quite natural because the inward flow is present at the dorsal side and the total material needs to be conserved. However, we did not find such a motion in the time-lapse images of actin. We have also tried to visualize the cytoplasmic flow using the GEMs particle of 40 nm radius. However, the Brownian fluctuation is too large to extract the average flow speed, which is quite small.

Therefore, the visualization of a three-dimensional flow profile is a future challenge.

5. The demonstration of actin filament motion within the concentric actomyosin ring using kymographs (Figure 4) is not very clear. It would be helpful if the authors could use a montage to show the motion of representative actin filaments, illustrating their rotational and centripetal movements.

We are grateful for the reviewer’s helpful comment on the presentation of the motion of actin filaments and have incorporated the suggested changes into the manuscript. We added snapshot images along the ventral and dorsal sides in Figure 5BC in the revised manuscript, showing that the actin fibers at the ventral side are almost immobile, while the actin fiber at the dorsal side moves clockwise as well as centripetally.

6. Concerning the origin of the concentric actomyosin rings, will these structures be disrupted when cells are treated with both Arp2/3 and formin inhibitors? If so, how will this affect the motion of the nucleus?

As shown in Figure 1, we have not observed any disruption of nuclear rotation when cells were treated with either Arp2/3 or formin inhibitors alone. When both inhibitors are used simultaneously, the concentric pattern of actin was maintained (Author response image 1), and the nucleus rotate with the speed comparable to the control cells (DMSO) (Author response image 1).

**Author response image 1. sa2fig1:** 

7. The authors should discuss how bundling of actin filaments in the concentric actin rings by actin crosslinking proteins might influence the model prediction of the chiral flow. In addition, the authors should demonstrate the sensitivity of the model to the choice of parameter values.

In the present model, based on our experimental results, the orientational field **p** is given as a concentric pattern at the dorsal membrane. Within the assumption of this model, the effect of crosslinking protein should be reflected in the parameter values, such as η, ζ^!^, ζ^"^. The present model is linear with respect to the variables and the orientational field **p**. Therefore, it is less sensitive to the choice of parameter values.

In the revised manuscript, we determine the parameter values to reproduce the experimental observation quantitatively. We first obtained the non-dimensional form of the equation of motion, which has only one parameter ζ given by Equation (10). As long as ζ is positive the rotational direction is clockwise when we view the cells from above. Since the myosin II induces the right-screwed motion of actin as shown in Figure 7A, ζ is positive. In this respect, the mechanism is robust against the parameter variation. We then determined the value of ζ so that the peak values of radial and angular speed were almost the same as shown in Figure 7—figure supplement 2. From these peak values, we can determine the velocity and time scales.

Consequently, the peak value in the angular speed of the cytoplasmic rotation agreed with the experimental observations (Figure 7H and Figure 6D). These procedures to constrain the parameter values from the experimental observation is explained in Material and Methods.

Minor points:1. The SMIFH2-treated cells show a higher rotation rate in the second five hours compared to the first five hours. Is there any difference in the actin organization during these time windows?

It is interesting to see whether actomyosin organization changes during this time interval in both control and SMIFH2-treated cells. However, to examine this point, we need to perform high-resolution time-lapse imaging for five hours using a sufficient number of cells. Because of time limitations, we consider this a future issue beyond the scope of the present manuscript.

2. It is worth mentioning that SMIFH2 has been shown to also inhibit several members of the myosin superfamily proteins, including on-muscle myosin 2A (PMID: 33589498).

In the original version of our manuscript, we have already discussed the inhibitory effect of SMIFH2 on myosin superfamily members, including myosin II, as this was reported in PMID: 33589498 (please see the second paragraph of the Discussion section). We have also noted that the same authors reported that SMIFH2 facilitates the centripetal movement of actin and myosin filaments when rat embryonic fibroblasts were used (PMID: 34455135), which is consistent with our observations using Caco-2 cells. In the case of Caco-2 cells, it is likely that the observed effects of SMIFH2 on actomyosin dynamics were not attributed to its potential inhibition of myosin II. Both effects have been mentioned in Discussion as below:

“Our experiments using an inhibitor and RNAi-mediated depletion have revealed that myosin II is involved in the chirality of Caco-2 cells, which is consistent with previous studies showing the involvement of myosin II in the chiral behaviors of several types of cells (Kumar et al. 2014). Interestingly, while some of these studies concluded that formins are essential for breaking the chiral symmetry (Tee et al. 2015-04, 2023; Davison et al. 2016; Kuroda et al. 2016; Abe and Kuroda 2019; Middelkoop et al. 2021), our results showed that the rotational speed of Caco-2 did not decrease but even slightly increased when treated with SMIFH2, a known inhibitor of formins, suggesting that formins are not required for Caco-2 cell chirality. Curiously, a previous study showed that SMIFH2 inhibits the centripetal movement of myosin II filaments in HFF (Nishimura, Shi, Zhang, et al. 2021), simultaneously demonstrating that this inhibitor also inhibits myosins, which apparently contradicts our observation. However, the same group also reported that SMIFH2 facilitated the centripetal movement of actin and myosin filaments when rat embryo fibroblasts were used (Nishimura, Shi, Li, et al. 2021), similar to our present observations as shown in (Figure 6 —figure supplement 1). These reports suggest that how cells respond to SMIFH2 depends on their types. In the case of Caco-2 cells, it is likely that the observed effects of SMIFH2 on actomyosin dynamics were not attributed to its potential inhibition of myosin II, although how this inhibitor induced the observed reorganization of actin and myosin filaments in Caco-2 cells remains unknown, as SMIFH2 seems to have multiple targets (Nishimura, Shi, Li, et al. 2021). Our experiments to deplete DIAPH2 and DAAM1 also support the idea that formins are not essential for breaking the chiral symmetry at least in the present cell system.”

3. From the example shown in Figure 4E and H, it seems that the centripetal movement of actin is faster in the control than in SMIFH2-treated cells. Is this the case?

We have shown in Figure 6—figure supplement 1 I in the revised manuscript that the centripetal movement of actin is faster in the SMIFH2-treated cells than in control cells. In the kymographs in Figure 5G and J of the revised manuscript, the flatter the line, the faster the movement.

4. Figure 4F-H is mentioned after Figure 5 in the main text, which is confusing.

We have rearranged the main text mentioning Figures 5 and 6 in the revised manuscript so that Figure 6 is now mentioned after Figure 5.

5. Arrow 3 in Figure 4D is not explained.

We have modified the text and figures so that all arrows shown in Figure 5 in the revised manuscript are now explained in the main text.

6. The cartoon diagrams in Figure 6E and their associated explanations lack clarity. Regarding Figure 6E middle: Why is the direction of the active torque depicted as counterclockwise? Concerning Figure 6E bottom: What is the orientation of the concentric actin ring in relation to the magenta rings (representing active torque)?Additionally, what does the black arrow at the ventral side of the diagram signify?

We acknowledge and agree with the reviewer’s comment that the diagram in Figure 6E in the original manuscript was not clear enough to understand. As for Figure 7E middle in the revised manuscript, the active torque, shown by the magenta arrow, is oriented in the direction of a right-handed screw pointing toward the center of the ring. Therefore, the magenta arrow is clockwise when we view it from the ring to the center direction. To see the arrow direction clearly, we have increased the arrowhead size in Figure 7E middle. We also clearly mentioned that the direction of active torque is clockwise as “the active torque is generated, which is the clockwise direction when viewed from the outside towards the center of the ring.”

Concerning Figure 7E bottom, this figure represents a side view of a cell. The orientation of the concentric actin ring is parallel to the horizontal axis. To enhance the clarity of this figure, we added an explanation in the figure legend, “A side view of a cell. The concentration of actomyosin increases in z (red color), leading to a gradient of active torque (magenta clockwise arrow) in the z direction, resulting in a rotational flow clockwise (black arrow).” The black arrow at the bottom indicates the direction of angle ϕ, which is the counterclockwise direction when viewed from above. To distinguish this black arrow from the above black arrows that show the flow direction, we have changed the color of this bottom black arrow to dark gray.

7. Figure 6—video 2: it is difficult to infer the rate of rotation by visualizing the movement of the white circle.

We have drawn a line within the white circle enclosing the nucleus to infer the rotation rate easily.

Significance:The presented work addresses a significant question in cell and tissue morphogenesis: how chirality can emerge at the single-cell level. The proposed model that cell-level chiral behavior can purely derive from the molecular chirality of myosin motor activity is intriguing. While the connection between the chiral behavior described in this work and tissue-level chirality remains to be demonstrated, the findings provide novel insights that may help in understanding the molecular origin of chiral asymmetry at the tissue and organ levels. The data acquisition, analysis, and presentation are of high quality, and the text is well-written and easy to follow. However, the study's limitation lies in the lack of direct evidence supporting the role of concentric actomyosin rings in generating the chiral flow. Additionally, the description of the characteristics of these concentric actomyosin rings requires further clarification.

We are grateful to the reviewer for acknowledging the importance of our work not only for cell chirality but also for chiral asymmetry at the tissue and organ level. We agree that more direct evidence is needed to support the role of the concentric actomyosin ring on the chiral rotating motion, which remains a future challenge.

Reviewer #2Major points:1. Overall, the authors provided an interesting model for chirality: myosin movement along the concentric actin fibers results in local active torque and a circular motion of cytoplasm, which further influence the nuclei rotation. This is, however, as the authors mentioned in the manuscript, contradictory to another publication where the transverse actin fiber is important in resulting chiral behavior in HFF cells. The author may want to address the differences and provide more evidence to explain this.

We fully agree with the reviewer’s point that our observation differs from the previous work on HFF. Indeed, we are discussing a new mechanism that has not been explored before. We have already mentioned in Discussion section of the original manuscript the differences of our observation from that using HFF as below:

“Previous studies using HFF proposed that radial actin fibers produce a force to drive the movement of the concentric actomyosin filaments (transverse fibers) through their connections (Tee et al., 2015). In the case of Caco-2 cells, however, we did not detect such radial fibers crossing the concentric actomyosin ring and our observations suggested that actomyosin ring by itself was moving in a clockwise direction through its own mechanism.”

2. From Figure 2, the increase of actomyosin after SMIFH2 seems promising, but the expected decrease of myosin in the Blebbistatin group seems not obvious. It also appears like an enhanced concentric pattern in Blebbistatin-treated cells (Figure 2C). If this treatment indeed leads to an increase of concentric patter of myosin, then based on the model proposed by the authors, the nuclei rotation should also be accelerated, which is not the case based on their quantification (Figure S4A). It would be more beneficial to provide a quantification showing how much concentric myosin is changed, and how much radial fibers is changed under different condition, so that the audience can have a better understanding of the model.

We are grateful for the reviewer’s comment regarding the actomyosin distribution in the cell treated with blebbistatin. First, the inhibition experiments of myosin II by blebbistatin or siRNA indicated that myosin II is the driving force for the chiral rotational motion. Therefore, rotational motion does not occur in cells treated with blebbistatin, regardless of how actomyosin is structured.

Our results indicate that the dorsally oriented actomyosin ring is the driving force for the chiral rotation, while the stress fibers at the ventral side showing a dextral swirling pattern do not contribute to the rotational motion. In the cells treated with blebbistatin, this chiral pattern of actin bundles disappeared at the ventral side. This is clearly seen in the ExM image of a cell treated with blebbistatin in Figure 4—figure supplement 1 in the revised manuscript, where the actin fibers at the ventral side did not exhibit a chiral pattern. In the main text, we now indicate this point more clearly as:

“Additionally, we examined cells treated with blebbistatin by ExM, confirming the results obtained by live imaging and conventional immunostaining (Figure 2A and Figure 3C). The chiral orientation of stress fibers was greatly reduced in the peripheral region after this treatment (Figure 4—figure Supplement 1). Furthermore, the dorsally located concentric actin filaments became undetectable in these specimens. These results are consistent with the idea that Myosin II plays a complex role, including in the organization of dorsal actin filaments.”

3. The difference between chiral actin stress fibers and concentric actin fibers is not clear. It would be more straightforward if a clear definition is provided. If the groups are solely determined based on thickness of the actin fiber, then a corresponding quantification showing clear size separation should be provided. The authors claim that concentric actin fibers is not chiral actin fibres, I wonder if there is a difference in length and density of these actin fibers. Consider a long and thin actin fiber is easier to curve (showing non-directional concentric pattern) compared to a thick and short actin fiber (showing directional chiral pattern). Also, is there a difference in terms of actin fiber orientation (barbed end and minus end orientation) among different groups of actin? Overall, I think the author need a better characterization and definition of the actin structure.

We thank this reviewer for commenting on the difference between stress fibers and concentric actin fibers. From the light-microscopy images, it is difficult to perform segmentation to identify individual actin fibers. Therefore, instead of grouping the two populations of actin based on their thickness, we distinguish between them based on their characteristics. One population is the “stress fibers,” located at the ventral side, anchored to focal adhesions at their termini and showing a dextral swirling pattern (red color in Figure 4A’ in the revised manuscript). The other population is called the “dorsal actin fibers” or “dorsal actomyosin fibers”, which are located along the dorsal membrane, and their orientations tend to be parallel to the tangential direction of the cell edge, forming a concentric pattern (green in Figures 4A’ and C’).

4. The drug treatments is influencing all different actin structures. Thus it is hard to have a clean and solid conclusion regarding the function of one group of actin structure. If, as the author proposed, concentric actin is important for nuclei rotation, then in the cells that lack radial fiber (or stress fiber), the nuclei rotation should not be affected. It would be interesting to deplete the anchor of radial fiber (either using vinculin mutant or laser ablation) and see whether the nuclei rotation is influenced in both wild type and SMIFH2 cases to further exclude that contribution from radial fibers.

We appreciate this comment. We have conducted experiments to deplete vinculin using its siRNA to see whether the stress fiber disruption influences the dorsal actomyosin and nuclear rotation. We found that the speed of the nuclear rotation was almost unchanged from the control experiment (see Figure 2—figure supplement 4 in the revised manuscript). Furthermore, some cells exhibited phenotype similar to SMIFH2-treated cells, where the stress fibers were disrupted and extended radially with a concentric pattern of actin fibers around the nucleus (Figure 2 —figure supplement 4 C and Figure 3B in the revised manuscript), suggesting that the ventral stress fiber is not essential for the nuclear rotation.

5. The CN03 experiment is hard to interpret, since it modified the localization pattern of both concentric actomyosin and radial actomyosin. Although it shows a cease on nuclei rotation, it is not clear what resulted in this phenotype. It might be beneficial to supplement this experiment with other experiments like using non-phosphorylatable myosin mutant cell, which may keep the pattern of actomyosin, but deplete the torque force.

We appreciate the reviewer’s comment regarding the experiment using Rho activator II (CN03). In the experiment using Rho activator II (CN03), we intended to clarify the role of dorsal actomyosin on the chiral nuclear rotation. Because it is not possible to introduce nonphosphorylatable myosin only to the dorsal side, we anticipate that the expression of this mutant myosin would result in the cessation of nuclear rotation, similar to the effect of blebbistatin, which inhibits chiral nuclear rotation (Figure 2A, B ,C in the revised manuscript). On the other hand, we agree that the experiment using CN03 does not explicitly reveal the causal relation between the dorsal actomyosin and the nuclear rotation. Therefore, we have softened the section title to “Depletion of dorsal actin and myosin coincides with the cessation of nuclear rotation.”

6. The local torque force model proposed by the author is interesting. The simulation in Figure 6 might be able to recapitulate in actual experiment. One potential way is to monitor subcellular organelle movement at different z-plane inside the cell, like a vesicle or free diffused lipid droplets (if any). The model might be more acceptable if the author can find an appropriate marker to trace the cytoplasmic flow at different zplane inside the cell to recapitulate the simulation.

We acknowledge the reviewer’s critical comment on the cytoplasmic flow profile.

We have tried to visualize the three-dimensional cytoplasmic flow using the GEMs particle of 40 nm radius. However, the Brownian fluctuation is too large to extract the average flow speed, which is quite small. Therefore, visualization of a three-dimensional flow profile remains a future challenge.

Minor points1. It would be better to provide movies/figures for myosin IIA/IIB RNAi cases to supplement the quantification.

In the revised supporting information, we provide a movie and figure of a cell treated with siRNA for myosin IIA and B in Figure 2—video 9, Figure 2—figure supplement 3 in the revised manuscript.

2. Figure 2A/3A wild type cells show a clear actin localization at the dorsal side of the cell, but in Figure 7A DMSO treated cell does not show a clear actin localization at the dorsal side of the cell.

Actin shows a localization along the dorsal membrane in Figure 7A in the previous manuscript (it is Figure 8A in the revised manuscript). We change the representation to see the localization more clearly.

3. The author may want to clearly label different groups of actin in at least one of the real images, not only in the cartoon, to help the audience to understand.

We acknowledge the reviewer’s valuable feedback on the presentation of the different groups of actin in the real images. The stress fibers are present on the ventral side, while the dorsal actomyosin fibers are located along the dorsal membrane. Therefore, we made a composite image where the ventral and dorsal cortex are distinguished in different colors (red and green, respectively) in Figure 4A’ and C’ in the revised manuscript.

4. Figure 5C and D, one of the three SMIFH2 cells show a similar trend as the wild type cell in terms of angular velocity, the authors may want to increase the data number.

We have increased the number of analyzed cells. According to the data we added, the SMIFH2-treated cells were not found to be faster in their rotational speed than the control cells.

As shown in Figure 1, in the second five hours of the ten-hour observation, the nuclear rotation speeds of the SMIFH2-treated cells tended to be maintained or slightly accelerated, while the control cells slightly decreased the rotating speed. Consequently, the SMIFH2-treated cells rotated faster than control cells on average. Therefore, to see clearly the difference in the cytoplasmic flow speed using LLSM, we should have controlled the observation time point after seeding cells for both control and SMIFH2-treated cells, but we did not. That could be one reason why the difference in the flow speed is not clearly seen. In this manuscript, we have decided not to discuss the difference in the cytoplasmic rotational speed nor pursue this further.

5. Figure S4B, the anti-GAPDH seems weak and hard to tell in the last four columns, so it is hard to believe the level of myoIIA/B is strongly reduced after RNAi treatment.

We acknowledge the reviewer’s comment regarding the unclear results of the western blot. We have repeated this experiment and obtained clearer data, as shown in Figure 2 figure—supplement 3 of the revised manuscript. We have also included immunostaining images of myosin IIA and myosin IIB in Figure 2—figure supplement 3 DC, showing that these proteins were depleted by siRNAs.

Significance:Overall, this manuscript proposed an interesting model where the chirality of the actinmyosin structure could result in a cytoplasmic flow that drives nuclear rotation without a clear cell-level chirality. Currently, the cellular and molecular mechanisms resulting in tissue and organ-level chirality remain less studied in the field. Although it used a simple single-cell model, this paper may provide some information to understand tissue chirality. It would be interesting to see whether this torque model can be applied at a multicellular scale.I am not an expert in mathematical simulation, so the rigor of the modeling simulation may require additional opinions. Cell biologists and developmental biologists interested in cytoskeleton and chirality may be influenced by the reported findings.

We would like to thank the reviewer for recognizing the importance of our study, which provides valuable information for the study of not only cell chirality but also the studies of tissue chirality and cytoskeleton in cell and developmental biology.

Reviewer #3 Major points1. The article emphasizes that molecular chirality governs chiral motion in the absence of a distinct cell-scale chiral organization of actin rings. This assertion implies that a macroscopic cell-scale chiral structure is necessary to induce cellular-scale chirality. This argument seems overly simplistic, as molecular chirality does not necessarily need to manifest as macroscopic chirality to be functional. Additionally, have the authors considered the possibility that the rings might actually organize into a chiral structure that has yet to be detected? For instance, could the rings form a spiral or helicoidal structure? Is the current microscopic resolution sufficient to rule out this possibility?

We acknowledge and agree with this reviewer’s argument that molecular chirality does not necessarily need to manifest as a macroscopic chiral structure for functional cell scale chirality. We have performed high-resolution microscopy, such as expansion microscopy (ExM) and lattice light sheet microscopy, but could not identify small-scale chiral structures in the dorsal concentric pattern.

2. In the Introduction, the authors appear to overlook part of the literature (see below) and some critical players in cell and organ chirality that are neither discussed nor tested in their system. This includes the conserved myosin1D, which play significant roles in chirality across species, first described in Drosophila (Speder et al. 2006; Hozumi et al. 2006) then in zebrafish (Juan et al. 2018), Xenopus (Tingler et al. 2018), and humans (Alsafwani et al. 2021; Yuan et al. 2024). Furthermore, the conserved role of formins, particularly diaphanous (Davison et al. 2016; Kuroda et al. 2016; Abe and Kuroda 2019) and DAAM (Chougule et al. 2020), is barely mentioned, despite their significance as common factors involved in chirality. The role of myo1D as a multiscale chiral factor exemplifies a clear case of a molecular determinant influencing chirality from the molecular level to the organismal level (Lebreton et al. 2018). The above literature should be cited and discussed within the context of this study. To strengthen the study, it would be beneficial to investigate the potential role of these conserved chiral factors (myo1D, dia, daam) within the Caco2 cell model. Testing these factors could provide a deeper understanding of their contributions to cellular chirality in this context.

We are grateful for the review’s comment on the literature, which should be cited in our paper. According to the suggestion, we have cited these previous works in the Introduction section but excluded Alsafwani et al. (2021) and Yuan et al. (2024), as we judge them as irrelevant to the present work. Concerning the potential role of myosin 1D, we have actually tested this possibility in the early stage of our study. However, we have not observed any significant effect of myosin 1D knockdown on the chiral nuclear rotation in Caco2 cells and, therefore, decided not to mention such negative results in our manuscript (Author response video 1). Below are the results of the analysis of angular velocity of nucleus of Caco-2 cells treated with siRNA for myosin 1D showing that the difference is not statistically significant and the western blot showing the proteins levels of myosin 1D in Caco-2 cells treated with the siRNA. For the myosin 1d depletion experiment, Silencer Select siRNA (MYO1D s9202) were used.

**Author response video 1. sa2video1:** 

For the formin knockdown experiments, see the next the answer to the next comment.

3. As noted by the authors, SMIFH2 is known for its lack of specificity in inhibiting formin function, as it also inhibits several members of the myosin family (Nishimura et al., 2021). Considering these limitations, the study would be more robust and convincing if the authors used specific inhibitors, such as siRNA, to target and invalidate formins (particularly diaphanous and DAAM).

According to the suggestion, we have conducted the depletion experiment of some of DIAPH and DAAM subfamilies to see their role on the rotating motion. We first studied the expression level of DIAPHs and DAAMs, and found that among them Diaph2 and DAAM1 were highly expressed (Figure2—figure supplement 2A). Therefore, we decided to deplete Diaph2 and DAAM1 by using siRNA. In Diaph2 knockdown cells, the distribution of actin fibers was similar to that in control cells, and the nuclear rotation persisted. In contrast, in DAAM1 knockdown cells, the actin bundle became oriented in a radial direction similar to SMIFH2-treated cells in some cells. However, this change in the actin organization did not prevent the rotation of nucleus in the clockwise direction. In conclusion, at least these formins are dispensable for the nuclear rotation. We show this result in the main text as below:

“These observations with formin inhibitor suggested that formins may not be essential for nuclear rotation. To further evaluate this possibility, we assessed the effect of knocking down formins using siRNA. Among the major mammalian formin family members (DIAPHs and DAAMs), DIAPH2 and DAAM1 were identified as being highly expressed in Caco-2 cells by RNA-sequencing (RNA-seq) analysis (Figure 2— figure Supplement 2A-C). In DIAPH2 knockdown cells, the distribution of actin fibers was similar to that in control cells, and the nuclear rotation persisted (Figure 2—figure Supplement 2DE and Figure 2—video 6). In contrast, in some DAAM1 knockdown cells, one end of the actin bundle appeared to be detached, resembling the phenotype observed in SMIFH2-treated cells (Figure 2—figure Supplement 2F and Figure 2— video 7). Notably, this change in the actin organization did not inhibit the clockwise nuclear rotation (Figure 2—figure Supplement 2D). These results support the idea that formins are not essential for the chiral rotation of the nucleus in Caco-2 cells, in contrast to previous observations (Tee et al., 2015; Davison et al., 2016; Kuroda et al., 2016;Abe and Kuroda, 2019; Middelkoop et al., 2021; Tee et al., 2023). Although we were not able to determine how SMIFH2 treatment induced F-actin reorganization, rather promoting nuclear rotation, as this reagent is not strictly specific to formins (Nishimura et al., 2021b), we used this inhibitor as a tool to investigate the mechanism of chiral rotational motion in further analysis.”

4. Figure 7DE: Could the restricted motion of the nucleus be a consequence of additional ventral F-actin, through binding and tethering?

We think that the ventral F-actin was increased by the CN03 treatment through binding and tethering as pointed out by this reviewer, which simultaneously led to a decrease of F-actin on the dorsal side. Due to this decrease in the dorsal F-actin, the driving force of actomyosin at the dorsal side to induce the nuclear rotation seems to have decreased. However, this experiment did not explicitly reveal the causal relation between dorsal actomyosin and nuclear rotation; we have softened the section title: “Depletion of dorsal actin and myosin coincides with the cessation of nuclear rotation.”

5. The authors mention that the cell membrane exhibits chiral movement. Could the authors provide further clarification on this finding? Specifically, does this imply that the entire cell rotates, or is the chiral movement confined to specific parts of the cell along the ventral-dorsal axis? Additionally, is this phenomenon observed in other models of cell chirality? If the entire cell undergoes rotation, it would suggest that (chiral) cell adhesion mechanisms may also be involved.

We are grateful for the reviewer’s comment on the motion of the entire cells and have addressed this problem by adding additional discussion in the Discussion section. The membrane on the dorsal side showed chiral rotations as shown in Video 2, while the stress fibers on the ventral side did not move as shown in the leftmost panel in Video 5 in the revised manuscript. Therefore, we consider that the dorsal part of the cell is twisting against the ventral part of the cell. Similar twisting behavior has been observed in zebrafish melanophores (Yamanaka and Kondo, 2014). Since the membrane behaves like a fluid, the twisting of the cell is likely to be resolved on the time scale of the cell rotation. We will include this consideration in Discussion of the revised manuscript as follows.

“Beads attached externally to the dorsal membrane exhibited both chiral rotation and centripetal movement, as shown in Video 2. This behavior mirrors the movement of actin filaments at the dorsal side, suggesting that the dorsal membrane moves in concert with the underlying actomyosin. In contrast, stress fibers located just above the ventral membrane did not move as shown in the leftmost panel of Video 5, suggesting that the ventral membrane is likely immobile. These observations imply that the dorsal region of the cell undergoes twisting relative to the ventral region. Given the fluid-like properties of membrane, such twisting is expected to be resolved over the time scale of cell rotation. Similar twisting behavior has been observed in zebrafish melanophores (Yamanaka and Kondo, 2015).”

Significance:By utilizing a specific cell line, the study offers rather incremental insights into the underlying mechanisms of cell chirality. It is unclear if the conclusions introduce any novel concepts to the field. The idea that molecular-scale chirality extends to larger scales is well established, with several examples documented in the literature (e.g., Tee et al., 2015, 2023; Chin et al., 2018; Lebreton et al., 2018).

We agree with the reviewer’s comment that the idea that cell- and organ-scale chirality arises from molecular-scale chirality is well recognized. However, how cell-scale chirality can be established remains unclear, although several ideas have been proposed. In our manuscript, we show a novel mechanism by which the molecular-scale chirality of actomyosin is converted to cell-scale chirality. We believe our finding is an essential contribution to the field, which seeks the mechanism for how molecular-scale chirality extends to larger scales.